# Robust trap effect in transition metal dichalcogenides for advanced multifunctional devices

Lei Yin[1,2,4], Peng He[1,2,4], Ruiqing Cheng [1,2,4], Feng Wang[1], Fengmei Wang[1], Zhenxing Wang[1,2], Yao Wen[1,2] & Jun He[1,2,3]

Defects play a crucial role in determining electric transport properties of two-dimensional transition metal dichalcogenides. In particular, defect-induced deep traps have been demonstrated to possess the ability to capture carriers. However, due to their poor stability and controllability, most studies focus on eliminating this trap effect, and little consideration was devoted to the applications of their inherent capabilities on electronics. Here, we report the realization of robust trap effect, which can capture carriers and store them steadily, in two-dimensional $MoS_{2x}Se_{2(1-x)}$ via synergistic effect of sulphur vacancies and isoelectronic selenium atoms. As a result, infrared detection with very high photoresponsivity ($2.4 \times 10^5$ A $W^{-1}$) and photoswitching ratio ($\sim 10^8$), as well as nonvolatile infrared memory with high program/erase ratio ($\sim 10^8$) and fast switching time, are achieved just based on an individual flake. This demonstration of defect engineering opens up an avenue for achieving high-performance infrared detector and memory.

[1] CAS Center for Excellence in Nanoscience, CAS Key Laboratory of Nanosystem and Hierarchical Fabrication, National Center for Nanoscience and Technology, Beijing 100190, China. [2] Center of Materials Science and Optoelectronics Engineering, University of Chinese Academy of Science, Beijing 100049, China. [3] School of Physics and Technology, Wuhan University, Wuhan 430072, China. [4] These authors contributed equally: Lei Yin, Peng He, Ruiqing Cheng. Correspondence and requests for materials should be addressed to J.H. (email: hej@nanoctr.cn)

The discovery of two-dimensional transition metal dichalcogenides (2D TMDs) provides an alternative choice to further scale down the device dimensions and an attractive platform for achieving new device functionalities[1–6]. Their atomic-scale thickness makes them very susceptible to intrinsic defects, enabling us to engineer their electronic and optoelectronic properties as desired[7]. So far, most studies of the deep-level defects in TMDs are concentrated on their suppression and healing[8–12]. This is because the in-gap deep levels always act as scattering or trapping centers, which are typically considered to be a limiting factor in device performance. However, these reported 'unfavorable' effects are only the tip of the iceberg, and plentiful defect types with specific functions remain unexplored. One highly desired defect function is programmable and stable trap effect, which possesses great potential in nonvolatile memory. To achieve this, using n-type semiconductor as an example, the defect-induced trap level ($E_t$) should be close to but below the Fermi level ($E_F$)[9], for it can capture and store minority carriers (holes, Fig. 1a) simultaneously with enhancing the tolerance to thermal perturbation. In addition, these trapped holes can be erased through recombining with the light-excited electrons from the valence band (Fig. 1b). With abundant anion vacancies, it has been demonstrated that deep defect levels are ubiquitous in TMDs[11–15]. However, the ideal trap effect has rarely been realized, implying that the properties of these defect levels need to be further modified.

Here, supported by density functional theory (DFT) calculations and electrical transport characterizations, we demonstrate experimentally that such a robust trap effect can be introduced in $MoS_{2x}Se_{2(1-x)}$ via the synergistic effect of S vacancy (SV) and substitutional Se atom. Most notably, the trap center's state of occupancy can be effectively controlled by external signals, namely programmed (holes are stored in trap levels) by gate

voltage pulses or erased (trapped holes are released) by laser pulses, allowing us not only to identify and maintain two distinctly different memory states but also to detect the laser with sub-bandgap wavelengths (e.g., 1550 and 1940 nm) at cryogenic temperature (80 K). The corresponding program/erase time is identified to be only ~0.3/0.9 ms. Consequently, our device, just based on an individual TMD flake, can simultaneously operate as a high-performance infrared detector and nonvolatile optoelectronic memory, providing a technical direction in simplifying fabrication procedures of multifunctional device and improving integration level.

## Results

**Physical model and DFT calculation.** Figure 1c shows the projected band structure of pristine $MoS_2$ with a single SV. The deep defect levels can be mainly attributed to the $4d$ states of Mo atom around the SV (more details are provided in Supplementary Fig. 1). It indicates that the property of defect levels is mainly determined by the local environment of SV, i.e., first-neighbor Mo atoms. Thus, to achieve the robust trap effect proposed above, we considered modifying the corresponding Mo-$4d$ orbital without inducing additional defect levels. As we know, isoelectronic chalcogen atoms (O, S, Se, and Te) own similar environment of valence electron. When S atoms around the SV were substituted with O, Se, or Te atoms, no new defect level was introduced in the bandgap (Supplementary Figs. 2–4). Moreover, compared with O and Te atoms, substitutional Se atoms can modulate the defect levels while maintaining the original electronic structure of $MoS_2$. This is due to the similar covalent radius and electronegativity of Se and S atom. Hence, for $MoS_2$ with SV, we adopt a rational modulation approach of isoelectronic substitution with Se atom.

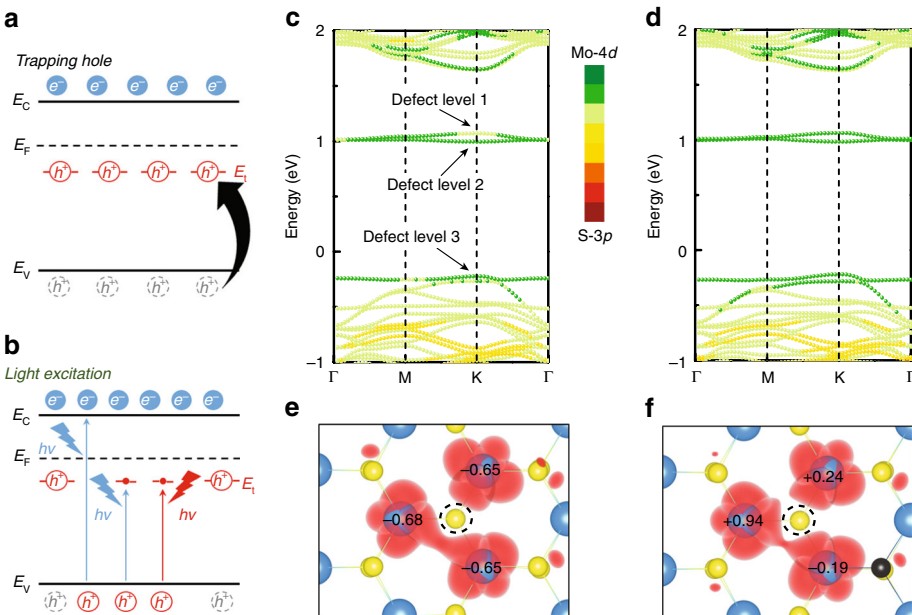

**Fig. 1** Physical model of carrier trapping. **a**, **b** Schematic band diagrams of deep traps ($E_t$) in a typical n-type semiconductor. Under non-equilibrium conditions, excess minority carriers (holes) will be captured and stored in $E_t$ (**a**). These trapped holes can be released through recombining with the light-excited electrons from valence band (**b**). Electrons and holes are represented by blue and red circles with $e-$ and $h^+$, respectively. Gray circles with $h^+$ are the disappeared holes. $E_C$, $E_V$, and $E_F$ indicate the conduction band edge, valence band edge and Fermi level, respectively. **c**, **d** Projected band structure of defective (with a single S vacancy) monolayer 2H-$MoS_2$ without (**c**) and with (**d**) a substitutional Se atom (corresponding Se concentration is 3.13%). Fermi level is set to zero. **e**, **f** Bader charge value (in unit of |e|) and wave function of defect levels in defective (with a single S vacancy) monolayer 2H-$MoS_2$ without (**e**) and with (**f**) a substitutional Se atom. Mo, S, and Se atoms are represented by blue, yellow, and black balls, respectively. S vacancy is marked by black dotted circle. The Bader charge value from negative to positive means that the ability to lose (obtain) electrons is gradually weakened (enhanced). The isosurface value of wave function is 0.1 e$Å^{-3}$. Source data are provided as a Source Data file

In the case of Se-doped $MoS_2$, the contribution of Mo-$4d$ states to the defect levels and valence band edge ($E_V$) is strengthened (Fig. 1d). Based on the Bader charge analysis (Fig. 1e, f, where positive/negative sign denotes the acquisition/loss of electrons, and the Bader charge value from negative to positive means that the ability to lose (obtain) electrons is gradually weakened (enhanced)), the Mo atoms around SV lose similar numbers of electron (0.66 |e| in average) due to the structural symmetry in non-doped $MoS_2$. However, after introducing a Se atom, the electron loss of Mo atom nearby the introduced Se atom decreases to 0.19 |e|. And the other two Mo atoms even begin to obtain electrons. It suggests that the ability to lose electrons of Mo atoms around SV is weakened by substituting S with Se atom.

This phenomenon is also confirmed by the wave function of defect levels at K point. As shown in Fig. 1e, f, the wave function overlap among the first-neighbor Mo atoms of SV is stronger than that of Se-doped one. It indicates that after introducing a Se atom, the electrons were localized at Mo atoms: that is, the electronegativity of Mo atoms is enhanced[11]. As a result, in the case of Se-doped $MoS_2$ with SV, the Mo atoms around SV could be a relative negative charge center in their local environment. The deep levels (DL-1 and DL-2) contributed by those Mo atoms are more likely to trap holes. On the contrary, the deep states in non-doped $MoS_2$ would serve as a donor state, which is consistent with the previous reports[10,13]. On account of above reasons, the introduction of Se atoms around SV can be regard as an efficient way to tune the property of in-gap defect states through modulating the $4d$ states of Mo atom. The formation energy of vacancy ($E^{form}$) and substitution by anions ($E^{sub}$) are discussed in Supplementary Figs. 5 and 6. The results indicate that (i) SV is easier to form in Se-doped $MoS_2$, and its formation location is independent of the location of the substitutional Se atom; (ii) the Se atom is more favorable to substitute S atom near SV. In addition, based on the evolution of band structure with increasing Se concentration ($C_{Se}$) from 0 to 25% (Supplementary Figs. 7 and 8), the almost unchanged energy different ($\Delta E_V$) between $E_t$ and $E_V$ is large enough to avert the effects of thermal perturbation on trapping holes.

**Anomalous temperature dependence of electronic transport.** To verify the proposed model, $MoS_{2x}Se_{2(1-x)}$ samples with Se concentration of 17–22% were chosen (Supplementary Figs. 9 and 10). Figure 2a shows the structure and atomic schematic of $MoS_{2x}Se_{2(1-x)}$-based device on $SiO_2$ (280 nm)/Si substrate (detailed manufacturing process is given in Methods). For consistency, the data in the main text was collected on one sample (labeled as device #1, flake thickness is 7.6 nm) unless otherwise noted. The corresponding optical microscopy (OM) and atomic force microscopy (AFM) images are displayed in Supplementary Fig. 11. First, its temperature-dependent electronic transport behavior was systematically studied. Figure 2b shows the $I_{DS}$–$V_{GS}$ transfer curves at temperatures from 300 to 80 K. Prominent n-type feature and excellent gate tunability with an ultrahigh on/off ratio of $10^9$ were observed at 300 K and maintained until the temperature drops to 160 K. However, when the temperature was below 140 K, the gate tunability starts to weaken, causing a negligible on/off ratio (the inset of Fig. 2b). This anomalous temperature dependence is opposite to the previously reported two-dimensional FETs[13,16–18], in which on/off ratio remains almost constant or increases with the decreasing temperature.

To elucidate the underlying mechanism of the anomalous temperature dependence, temperature-dependent conductivity and activation energy ($E_a$) are analyzed with the thermally activated transport model. As shown in Supplementary Fig. 12, the activation energy is dependent on gate voltage, and is lower

than that of other 2D materials on $SiO_2$ substrate[16,19]. Such low $E_a$ value suggests the Fermi level of channel material is very close to its mobility edge. Thus, the unoccupied defect levels upon $E_F$ are very shallow and difficult to trap electrons stably even under low temperature. By contrast, the deep defect levels below $E_F$ can trap and store holes more stably at low temperature due to the weak thermal perturbation. These trapped holes with positive charge can effectively screen the effect of negative gate voltage ($V_{GS}$) and suppress the depletion of electrons in the channel, generating the anomalous high-current state at negative $V_{GS}$ (Supplementary Fig. 13). This model (hole trapping and de-trapping) is further supported by the photoresponse measurement with infrared lasers and time-resolved current measurement under various temperature, which demonstrate the existence of in-gap traps and the influence of temperature on de-trapping process, respectively.

The effect of thermal perturbation on holes' de-trapping process is characterized by de-trapping time ($\tau_t$): $\tau_t^{-1} = s_p N_v v_{th} \exp(-\Delta E/(kT))$[9,20], where $s_p$, $N_v$, $v_{th}$, $\Delta E$, and $k$ are the capture cross section of the trap center, the effective density of states in the valence bands, the thermal velocity of the carriers, the energy difference between the trap state and the valence band edge, and the Boltzmann constant, respectively. For a certain trap state, with the decrease in temperature, it will be difficult to thermally excite the trapped holes back to the valence band due to the low thermal energy ($kT$). It suggests that the effect of thermal perturbation on holes' de-trapping process starts to weaken. And the de-trapping time increases with the decrease of temperature[21]. At this point, we can easily detect the anomalous electronic transport properties induced by the trapped holes. As shown in Fig. 2c, the current with a reading voltage of 1 V maintained very well at 80 K ($V_{GS} = -80$ V). However, it became unstable with the increasing temperatures, implying the thermally assisted de-trapping process. In this case, the holes' de-trapping process starts to weaken at 140 K (defined as the critical temperature, $T_C$), and is completely suppressed at 80 K. These results not only further solidify the above discussed physical model, but also help determine the optimum operating temperature of the device.

The Raman spectra show that there is neither characteristic peak shift nor new peaks in the process of changing temperature, indicating no generation of structural phase transition (Supplementary Fig. 14). In addition, a number of devices with various thicknesses were fabricated on both $SiO_2$ and h-BN substrates (Supplementary Figs. 15 and 16). The corresponding temperature-dependent on/off ratios and $T_C$ were extracted and are summarized in Fig. 2d. The leakage currents of devices at various temperature are also provided in Supplementary Figs. 17 and 18. The good repeatability of anomalous gate tunability and negligible leakage currents eliminate the possible effects from substrates. It is noteworthy that although the hole trapping is introduced by isoelectronic substitution, the transistors still exhibit excellent switching characteristics and fast carrier transport. Figure 2e shows the statistical carrier mobilities at room temperature, which is compared to that of most back-gated FETs based on TMDs[22–24].

**Infrared photoresponse properties.** In this section, we demonstrate experimentally the existence of in-gap trap levels and their application in infrared photodetection. Figure 3a shows the $I_{DS}$–$V_{GS}$ transfer curves under illumination with 473, 1550, and 1940 nm lasers at 80 K, where negative photoresponse (illumination current is lower than dark current) was observed under negative $V_{GS}$ region. This behavior is attributed to the fact that electrons are photo-excited from the valence band onto the trap levels and

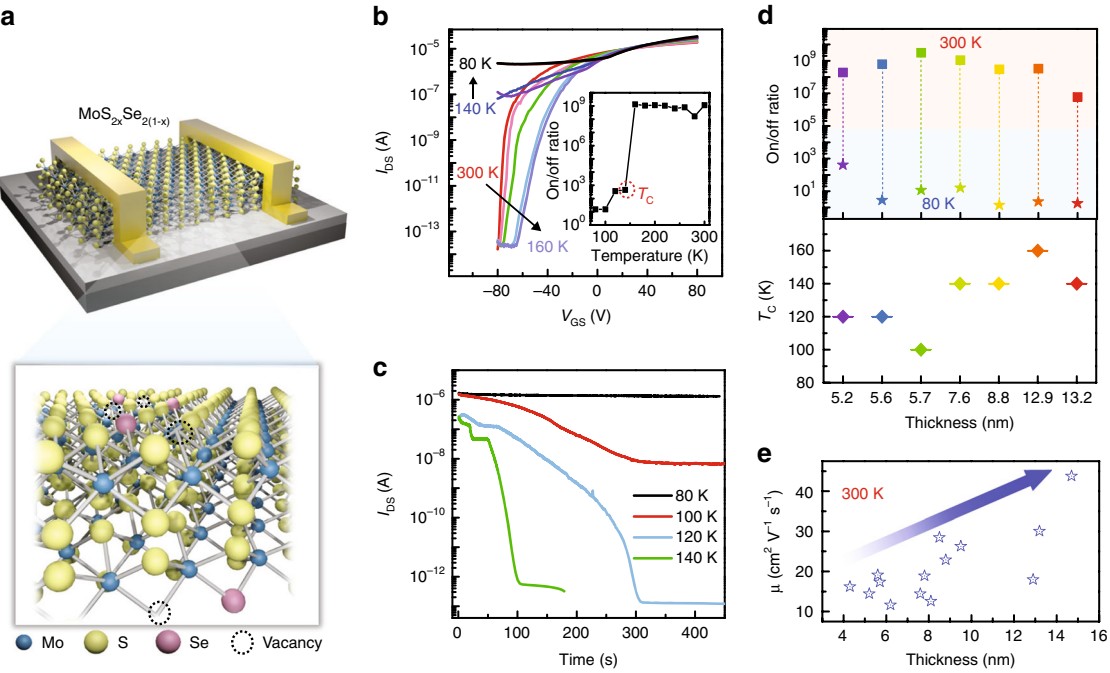

**Fig. 2** Electrical transport properties of the device. **a** Structure and atomic schematic of the device. S vacancies and isoelectronic Se atoms synergistically serve to introduce expected trap centers in $MoS_{2x}Se_{2(1-x)}$. **b** $I_{DS}$–$V_{GS}$ curves under various temperatures. $V_{DS} = 1$ V. Inset, plot of the current on/off ratios along with temperatures. The corresponding critical temperature ($T_C$), at which the on/off ratio drops, is indicated by the red circle. **c** Time-resolved current measurement at $V_{GS} = -80$ V and $V_{DS} = 1$ V. **d** The statistics of temperature-dependent on/off ratios (upper panel) and $T_C$ (lower panel) of a number of devices with various thicknesses. **e** The statistics of room temperature field-effect mobilities (μ) obtained from 13 devices. Source data are provided as a Source Data file

then recombine with the trapped holes. The corresponding band diagrams are given in Supplementary Fig. 19. As a result, electrons in the conduction band are depleted by negative $V_{GS}$ due to the vanishing charge shielding effect. Compared with 473 nm laser, the negative photoresponse under infrared lasers is more prominent. Similar behavior is also reflected in the $I_{DS}$–$V_{DS}$ output curves (Supplementary Fig. 20). This is because the high-energy photons (2.62 eV) of 473 nm laser can excite electrons not only to the trap levels but also to the conduction band, which can generate a positive photocurrent contribution. In other words, it is a combined effect of interband and band-edge light absorption.

This process is further confirmed by time-resolved photo-response measurement at $V_{GS} = -80$ V. Under the first illumination with 473 nm laser, the current dropped immediately and remained at a constant value (~$10^{-9}$ A, blue line in Fig. 3b). In this scenario, part of the photo-excited electrons is recombined with the trapped holes to eliminate charge shielding effect, and the other part contributes to the photocurrent. When the laser was switched off, the current reduced again to the order of ~$10^{-13}$ A, implying that the trapped holes have been recombined with the photo-excited electrons. Subsequently, the device exhibits a normal photoswitching behavior (i.e., positive photoresponse) with the switch of 473 nm laser. However, the low-energy photons of 1550 nm (0.8 eV) and 1940 nm (0.64 eV) lasers cannot pump electrons to conduction band. Thus, when turning on the laser, the current reduced directly to the order of ~$10^{-13}$ A (pink and red line in Fig. 3b). And the current kept at that value even the laser was turned on again. Particularly, the negative photoresponse appears only once under all three different illumination conditions, suggesting that holes cannot be captured again at $V_{GS} = -80$ V due to the electrostatic attraction between the large negative gate voltage and positive charges. Thus, in the following several photoswitching cycles, the

defect levels are in the occupied state (without trapped holes), and electrons in the valence band could not be excited to these defect levels. To realize repeatable photoresponse properties, a suitable $V_{GS}$ pulse should be applied to capture holes. The gate voltage-dependent carrier capture will be discussed in the following section. Figure 3c shows the multiple electro- and photo-excitation cycles. With a +80 V gate voltage pulse, holes were trapped into the defect levels, leading to the appearance of high-current state at $V_{GS} = -80$ V. After applying a laser pulse, the current returns to the low state. Transient response time of the device was measured by an oscilloscope method (see Methods section), and a full cycle of measurement is displayed in Supplementary Fig. 21. As shown in Fig. 3d, the partial enlarged view of the falling (rising) edge region triggered by voltage (1550 nm laser) pulse illustrates that the response time of electro- (photo-) excitation is about 0.3 (0.9) ms.

The power density (P)-dependent photoresponse was also studied (Supplementary Fig. 22). As for 473 nm laser, the illumination current ($I_{illumination}$) at negative $V_{GS}$ region reduces more conspicuously as the power density decreases, resulting in a larger photocurrent ($|I_{ph}|$). Here, $I_{ph}$ is the difference between illumination current and dark current ($I_{dark}$). It implies after optically erasing the trapped holes, the device exhibits a normal $I_{illumination}$ that weakens with the decrease of laser power density (like band-edge light absorption of other 2D photodetectors). However, the illumination current does not change with the power intensity for 1550 and 1940 nm lasers. It is reasonable to anticipate that the number of excited electrons is already more than the trapped holes (i.e., the photoresponse reaches saturation) within our experimental conditions. And when applying a much lower power density, it is possible that the trapped holes won't be erased completely, thus generating a power density-dependent photoresponse. These phenomena further confirm that the

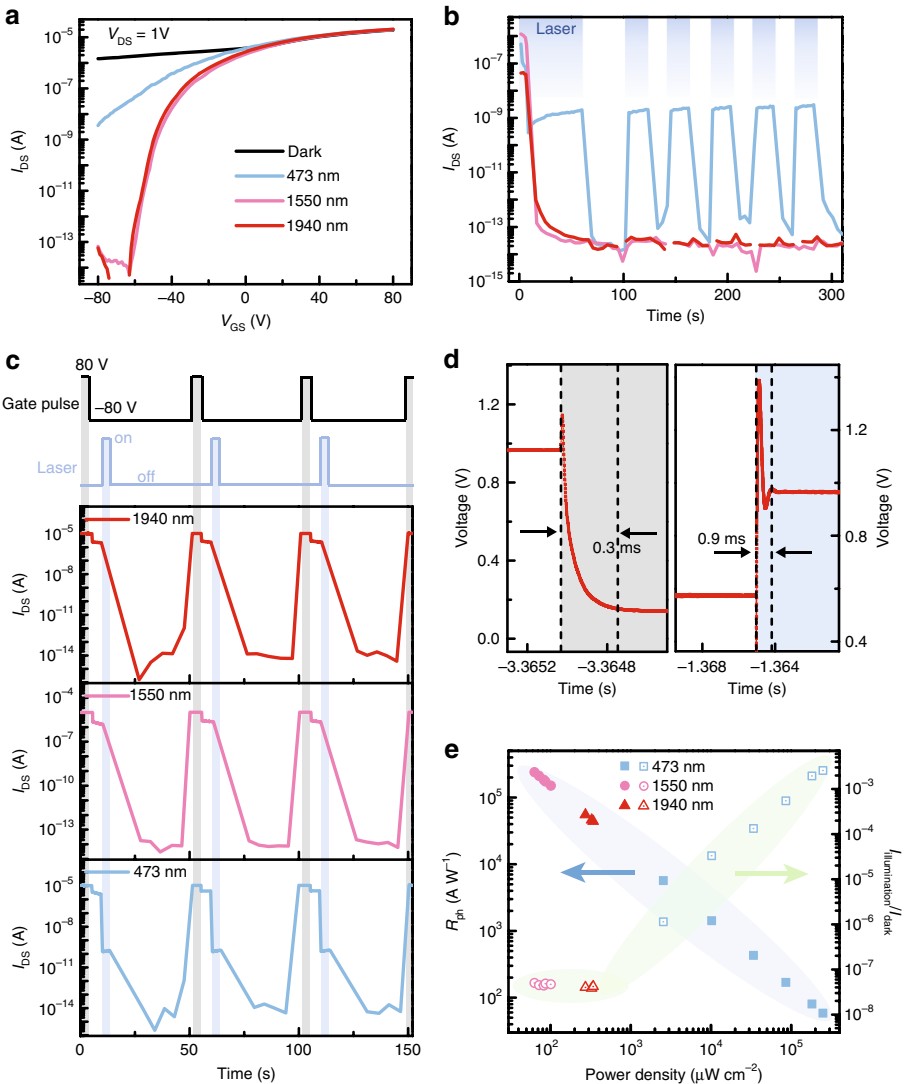

**Fig. 3** Photoresponse properties of the device. **a** $I_{DS}$–$V_{GS}$ curves of the device under dark and illuminated states with 473 nm (248 mW cm$^{-2}$), 1550 nm (100.6 μW cm$^{-2}$) and 1940 nm (327.1 μW cm$^{-2}$) lasers. $V_{DS} = 1$ V, $T = 80$ K. **b** Time-resolved photoresponse measurement at $V_{GS} = -80$ V. The figure legend is same as (**a**). The blue rectangle represents the illuminated state. **c** Multiple electro- and photo-excitation cycles with the gate voltage pulses (+80 V, 8 s) and laser pulses (5 s). **d** Response time of the electro- (left panel) and photo-excitation (right panel) process. The laser wavelength is 1550 nm. **e** The corresponding photoresponsivity ($R_{ph}$) and $I_{illmination}/I_{dark}$ ratios as a function of power density at $V_{GS} = -80$ V. Source data are provided as a Source Data file

infrared photoresponse originates from the in-gap trap levels, whereas the photoresponse with 473 nm laser is a result of the combined effect of interband and band-edge light absorption. To evaluate the photoresponse performance of our device, its photoswitching ratio ($I_{illmination}/I_{dark}$) and photoresponsivity ($R_{ph} = |I_{ph}|/(PS)$[25], $S$ is the device active area) were extracted and are summarized in Fig. 3e. The ultrahigh $I_{illmination}/I_{dark}$ of $10^8$ was achieved for infrared illumination, and the largest $R_{ph}$ reached $2.4 \times 10^5$ and $5.5 \times 10^4$ A W$^{-1}$ for 1550 and 1940 nm laser, respectively. Other important parameters (photocurrent and gain) were also calculated and displayed in Supplementary Fig. 23. The performance of our infrared photodetectors is comparable to the reported infrared photodetectors based on 2D materials (Supplementary Table 1). It's worth noting that when the photon energy exceeds 0.64 eV, the apparent negative photoresponse can always be observed (Supplementary Fig. 22). However, no obvious photoresponse was observed under 2699 nm (0.46 eV) and 3000 nm (0.41 eV) lasers (Supplementary

Fig. 24). Therefore, we estimate that trap distribution should mainly concentrate in the range of 0.46–0.64 eV above the $E_V$. Detailed photoelectric characteristics of another device (labeled as device #2, flake thickness is 13.2 nm) were given in Supplementary Fig. 25.

**Infrared memory and its operational mechanism.** Optoelectronic memory, especially infrared memory, has attracted great attention for its wide applications in long-distance secure communication, information capturing and processing[26–30]. However, the current reported optoelectronic memories need the aid of additional charge transfer/storage medium (such as heterostacks[28,29], floating gate[31,32], and functional substrates[33,34]). Here, utilizing the intrinsic trap centers of channel material, a simple but high-performance nonvolatile optoelectronic memory was demonstrated. As shown in Fig. 3a, a maximum difference ($10^8$) between the ON (dark) and OFF (infrared illuminated)

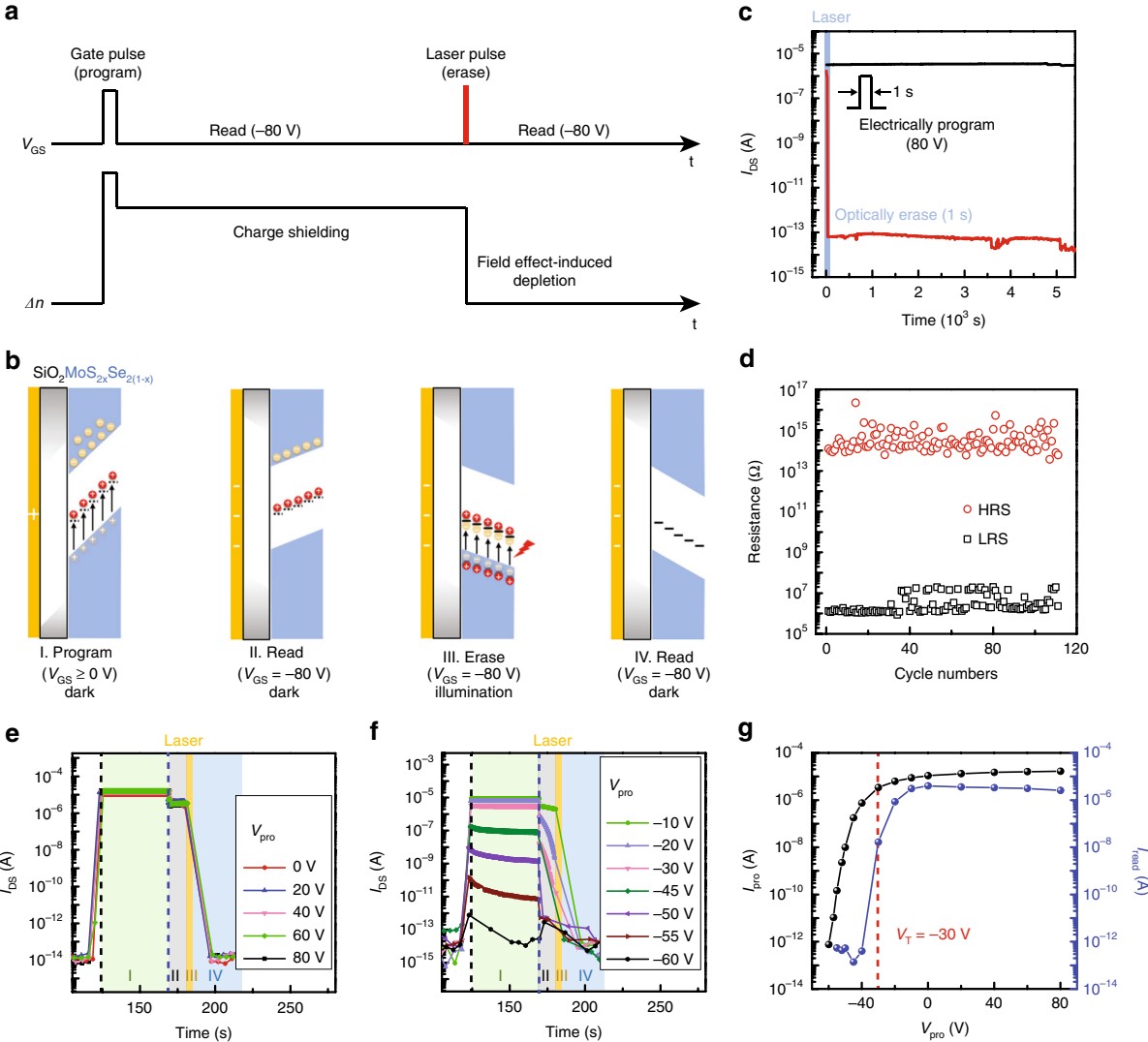

**Fig. 4** Gate voltage-dependent memory effect and performance evaluation. **a** Upper panel, schematic illustration of the program, readout and erase process of a single operation cycle. Lower panel, the corresponding time evolution of electron concentration ($\Delta n$) in device channel. **b** Schematic band diagrams of the corresponding program (I), readout (II, IV), and erase (III) states. Electrons and holes are represented by yellow and red balls. Gray balls are the disappeared electrons or holes. Black dashed and solid lines in the bandgap denote the empty state (with trapped holes) and occupied state (without trapped holes). **c** Retention performance of two memory states after electrically programming and optically erasing. **d** Endurance characteristics of the memory for over 100 program/erase cycles. Laser wavelength and power density are 1550 nm and 100.6 $\mu$W cm$^{-2}$, respectively. **e, f** Time evolution of current in a single operation cycle with various programming voltages ($V_{pro}$). $V_{DS} = 1$ V, $T = 80$ K. Region I, II, III, and IV denote the states of programming, readout (under program state), erasing, and readout (under erase state) shown in (**b**), respectively. **g** The plot of programming and read current ($I_{pro}$ and $I_{read}$, the current values along the black and blue dashed line in **e, f**) versus $V_{pro}$. Threshold voltage ($V_T$) is a critical gate voltage for storing charges. Source data are provided as a Source Data file

states was observed under $V_{GS} = -80$ V and $T = 80$ K, implying two distinctly different memory states.

The schematic diagram of a single operation cycle, involving programming, readout, and erasing processes, is shown in Fig. 4a and b. The programming operation is achieved by applying a nonnegative gate voltage ($V_{GS} \geq 0$ V) to the devices. In this scenario, excess holes are trapped into the defect levels (State I in Fig. 4b). At the same time, non-equilibrium electron concentration ($\Delta n$) increases due to the electrostatic electron-doping. These stored holes can be retained even after removing the corresponding positive $V_{GS}$, thereby acting as positive charge centers localized in the bandgap (State II). Benefiting from their shielding effect to negative gate field (suppress the depletion of electrons in the channel), the semiconductor band at semiconductor–insulator interface cannot change dramatically and a retentive low resistance

state (LRS) can be readout at $V_{GS} = -80$ V. To erase the stored holes, infrared laser pulse is applied under $V_{GS} = -80$ V (State III). The photo-excited electrons can recombine with the stored holes to eliminate the charge shielding effect. As a consequence, electrons upon the conduction band will be depleted by the large negative gate voltage, and the device goes back to a retentive high resistance state (HRS, State IV). The entire testing process is shown in Fig. 4e.

The stability of charge storage and operation endurance are important evaluation parameters for memory performance. Figure 4c displays the retention performance of two memory states after electrically programming with gate voltage pulse ($+80$ V, 1 s) and optically erasing with laser pulse (1550 nm, 1 s). 1550 nm is a typical communication waveband, which can minimize energy losses during transmission. The current shows no noticeable change over $5 \times 10^3$ s, indicating that using this

measurement condition will not disturb its retention characteristics. In addition, as demonstrated in Fig. 4d, our device exhibits highly repeatable switching behaviors between LRS and HRS over 100 cycles. Even testing again after a year, the device still maintains its stability and high program/erase ratio ($\sim 10^7$) within 600 operation cycles. The test method and other experimental results with 473 and 1940 nm lasers are given in Supplementary Fig. 26. Moreover, the high program/erase current ratio and fast switching time (Fig. 3e), which are superior to the reported optoelectronic memories[28,34,35] (Supplementary Table 2), ensure the accurate information identification and high-speed data processing. The wavelength dependence of erasing speed was also studied (Supplementary Fig. 27), which further confirmed the proposed erasing mechanism of interband light absorption. And the power consumptions during the electricity (for programming) and laser (for erasing) actuation are 3.5 pJ and 8.4 fJ (1550 nm)/25 fJ (1940 nm), respectively. Such fast operation speed and low power consumption may be attributed to the simple device configuration. Unlike the heterostructure-based nonvolatile optoelectronic memories, our devices are just based on an individual TMD flake. Thus, in the programming and erasing processes, there is no need for charge carriers to transfer across the hetero-interface. To further improve the operation speed of our devices, adopting optimized contact metal or high-$k$ dielectric materials could be an effective strategy.

Next, we systematically studied the gate voltage-dependent storage characteristic that involves electric field effect and trap effect. First, due to electrostatic electron-/hole-doping, $\Delta n$ changed when different programming $V_{GS}$ ($V_{pro}$) were applied, generating various output currents (green shadow region of Fig. 4e, f). The current values along the black dashed line in Fig. 4e, f are defined as programming current ($I_{pro}$) and extracted as the black curve in Fig. 4g. Therefore, a transfer curve-like profile is obtained, revealing the field effect feature. Second, the ability of trapping holes can be tuned by $V_{pro}$, leading to different readout states and relaxation time (region II in Fig. 4e, f). This high sensitivity of readout states to $V_{pro}$ pulse implies its potential in emulating a long-term plasticity of synapse. For a better understanding, the read current ($I_{read}$, the current values along the blue dashed line in Fig. 4e, f) was extracted as a function of $V_{pro}$ (the blue curve in Fig. 4g). And the schematic diagrams of various program processes and the corresponding read processes are displayed in Supplementary Fig. 28. When $V_{pro}$ is set in the range of $-80$ to $-30$ V, the strong electrostatic attraction between the large negative gate voltage and positive charges is not conducive to capture and store holes. Thus, in the absence of charge shielding centers, the electrons in the channel are depleted by large negative gate voltage, and the LRS (measured at $V_{GS} = -80$ V, $V_{DS} = 1$ V) cannot be effectively identified. Along with the increasing $V_{pro}$ ($-30$ V $< V_{pro} < 0$), the LRS can be easily readout and gradually becomes stable. It suggests that holes start to be captured by defect levels, which can provide stronger charge shielding effect to the gate field of $-80$ V. Hence, $V_{pro} = -30$ V is a threshold voltage ($V_T$) for storing charges here. Further, when a nonnegative $V_{pro}$ is applied, the corresponding electrostatic attraction disappears, and the capture of holes becomes much easier. Sufficient numbers of trapped holes give rise to a robust charge shielding effect. Thus, very stable LRS can be obtained at $V_{GS} = -80$ V. Detailed characterizations of device #2 were given in Supplementary Fig. 29. Considering the fact that the robust trap effect is independent of the substrate, the device operation voltage can be effectively reduced by introducing high-$k$ dielectric layer.

## Discussion

In summary, we have demonstrated that the stable trap effect can be introduced in $MoS_{2x}Se_{2(1-x)}$ nanosheet via the modulation of isoelectronic Se atoms on the defect levels induced by SV, and can be effectively tuned by external gate voltage pulse or laser pulse. With the aid of this robust trap effect, our device can operate as an infrared detector with high photoresponsivity ($2.4 \times 10^5$ A W$^{-1}$) and photoswitching ratio ($\sim 10^8$), and also can operate as nonvolatile optoelectronic memory with high program/erase ratio ($\sim 10^8$) and fast switching time ($\tau_{electro} = 0.3$ ms, $\tau_{photo} = 0.9$ ms). This study could not only deepen the understanding about the defect functions in TMDs, but also open a route for designing advanced multifunctional devices.

## Methods

**DFT calculations**. All density function theory (DFT) calculations for pristine and defective MoS₂ were carried out by using the projector-augment wave method with the Vienna ab initio simulation package (VASP) code[36,37]. The generalized gradient approximation (GGA) of the exchange and correlation functional was described by Perdew-Burke-Ernzerhof (PBE)[38]. A plane-wave cutoff energy of 500 eV and a $5 \times 5 \times 1$ Gamma-center K-point mesh is used to relax all slab models. The MoS₂ models with a single SV were constructed from two different kinds of supercells: a ($2 \times 2$) supercell and a ($4 \times 4$) supercell (Supplementary Figs. 2 and 5). They are used to study the substitution influence of various isoelectronic chalcogen atom (O, Se, and Te) and Se concentration on the band structure of pristine MoS₂, respectively. Both the locations of SV and substituted O, Se, and Te atoms are on the upper surface of MoS₂. Each simulation cell includes ~15 Å of vacuum to separate the periodic image along the $z$ direction. All atoms are relaxed until the force on each atom is <0.01 eV Å$^{-1}$. The Bader charge were calculated using the algorithm described in ref.[39]. The band structures were calculated using PBE functional to be consistent with other calculations[13]. The hybrid functional (HSE06), which can more accurately describe the bandgap, was also used (Supplementary Fig. 4).

For the case of Se-doped MoS₂, each calculated cell consists of 16 anionic sites (Mo atoms) and 32 anionic sites (one SV, $x$ Se atoms, and $31-x$ S atoms). To facilitate discussion, we assume that the concentration of SV is unchanged during modulation. The S vacancy (doped Se) concentration is defined as the ratio of the number of S vacancies (doped Se atoms) and total anion coordination number. Hence, this allows us to have increments of 3.13% for one more doped Se atom. For instance, four doped Se atoms correspond to Se concentration of 12.5%. All subsequent doped Se atoms from the first one is placed as near as possible to the existing S vacancy. Because of the similar covalent radius of S and Se atom, the lattice of Se-doped MoS₂ is almost unchanged. Therefore, we fixed the lattice parameter of doped MoS₂ at 3.13 Å, which is equal to that of pristine MoS₂.

**Material characterization**. The $MoS_{2x}Se_{2(1-x)}$ flakes were obtained by mechanically exfoliating bulk crystal (2D semiconductor). Their crystallinity quality and chemical composition were determined by transmission electron microscopy (TEM) combined with energy-dispersive X-ray spectroscopy (EDX), atomic-level high-angle annular dark-field scanning transmission electron microscopy (HAADF-STEM), and X-ray photoelectron spectroscopy (XPS), respectively. Raman spectra were measured using a confocal microscope-based Raman spectrometer (Renishaw InVia) with an excitation laser line of 532 nm.

**Device fabrication and characterization**. First, few-layer $MoS_{2x}Se_{2(1-x)}$ were mechanically peeled and transferred onto 280 nm SiO₂/Si wafers. The Si substrates are n-doped with phosphorus and its resistivity is about 1–30 Ω·cm. And the oxide on the substrate is thermally grown. The substrates were cleaned before use, with a mixed solution of H₂O₂ and H₂SO₄ (volume ratio is 3:7) for 2 h. Optical microscopy (Olympus BX51M) and atomic force microscopy (Dimension 3100) were used to select the exfoliated flakes and confirm thickness of them. The source/drain electrodes (10/50 nm Cr/Au) were fabricated by standard electron-beam lithography (Nova200 NanoLab) and thermal evaporation apparatus (with a rate of 0.05 Å s$^{-1}$). Finally, a standard lift-off process was adopted.

A probe station (Lakeshore, TTP4) equipped with a vacuum pump, a flow cryostat, and a semiconductor characterization system (Keithley 4200) was used to measure electrical transport of devices. The optoelectronic properties of devices were characterized under illumination of 473, 808, 1550, 1940, 2699, and 3000 nm. The spot diameter of laser is 0.3, 0.8, 1.07, 0.5, 1.5, and 1.5 cm, respectively. The laser pulse and gate voltage pulse were produced by a chopper and a wave generator, respectively.

The real transient response time was measured by an oscilloscope method, where our device was connected in series with a 1 MΩ resistor. When the device is excited by external signals, an oscilloscope is used to record the bias change of the device, to probe the current variation of the series circuit. The response time is defined as the time interval between two steady states.

## Data availability

The source data underlying Figs. 1c, d, 2b–e, 3a–e, and 4c–g and Supplementary Figs. 1, 3, 4, 8–12, 14–18, 20, 22–26, and 29 are provided as a Source Data file. All data are also available from the corresponding author upon reasonable request.

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

## Acknowledgements

This work was supported by Ministry of Science and Technology of China (No. 2016YFA0200700), National Natural Science Foundation of China (Nos. 61625401, 61474033, 61574050, 11674072, and 21703047), Strategic Priority Research Program of the Chinese Academy of Sciences (Grant No. XDA09040201), and CAS Key Laboratory of Nanosystem and Hierarchical Fabrication. The authors also gratefully acknowledge the support of Youth Innovation Promotion Association CAS.

## Author contributions

J.H. conceived and supervised the project. L.Y. and R.C. fabricated the devices and performed electrical and optoelectronic measurements. L.Y. carried out the Raman and AFM measurements. P.H. performed density function theory calculations. L.Y., R.C., and J.H. analyzed the data and co-wrote the paper in consultation with P.H., F.W., F.M.W., Z.W., and Y.W.

## Additional information

**Competing interests:** The authors declare no competing interests.

