## [Peer Review File · Nature Communications]

Reviewers' comments:

Reviewer #1 (Remarks to the Author):

In the manuscript by Lei Yin et. al. titled "Robust trap effect in transition metal dichalcogenides for advanced multifunctional devices" its authors report the realization of robust defect-induced trap effect to construct the multifunctional electronic and and optoelectronic devices. The authors discuss both the physical model and the electrical transport properties of the device in detail. The authors then show that the device shows infrared detection with very high photoresponsivity and high photoswitching ratio. Finally, they use the device to demonstrate the nonvolatile memory properties with high program/erase ratio and fast optical erasing time. The results are fairly impressive and are a fit for this journal. I recommend it be accepted after addressing the following issues:

1. In Fig. 2, there are too many description about the temperature dependent properties of the device. However, these properties seems to have little relevance to the article. Could the authors add some explanation about the relationship between the temperature dependent properties and the multifunctional properties of the device.
2. In the "Infrared memory and its operational mechanism" section, the authors use the electrical pulse to programme the device, and use the laser pulse to erase device. However, in normal memory device, people always use one kind of physical way to induce both programme and erase behavior. Could the authors demonstrate that your device can be programmed/erased only using the electrical pulse or laser pulse.
3. Can the authors draw some schematic diagrams to explain the mechanisms of the device during the gate voltage-dependent memory operation?
4. Further discussion needs to be added for power consumption of these devices during the gate voltage-dependent memory operation.
5. For tangible operation, the device should work at the level of tens of ms. Please compare the operation speed with previous reports. Do you have suggestion to improve the operation speed?
6. There are some reports about the nonvolatile operation of 2-d materials based device? What is the difference between your device and the other devices reported by other groups.
7. In my opinion, your device with the nonvolatile memory and the photoelectricity properties is not novelty. However, the application of using these properties to demonstrated some new application would be interesting and novel. For example, these properties can be used to demonstrate some neuromorphic properties (memristor or synaptic emulation). Could the author give some discussions or prospects about the new application of the device?
8. There are several grammatical and typos which make reading difficult. They need to be dealt with before this article is published.

Reviewer #2 (Remarks to the Author):

The manuscript by Yin et al. represents a report on advanced multifunctional devices based on metal dichalcogenides using defect/substitution induced traps. The authors explore a new way to use defects in their advantage contrasting to most of the device aspects where defects are attributed to the detrimental of the device performance. They have shown that using stable hole traps, both infrared photodetector as well as nonvolatile memory with reasonably high performance can be obtained. The detailed and elaborative device study along with DFT prediction for the trap formation is shown.

The device performance and the optoelectronic engineering of metal dichalcogenides sheet seems novel and interesting. However, several physical aspects of the device performance as well as the results claimed in the manuscripts are missing or unclear. In this present form, the manuscript cannot be published for a journal like Nature Communications. However, if the authors can address the following concerns and incorporate them in the manuscript, it can be considered for publication.

- (1) The authors discussed DFT results obtained with PBE functional, which is well known to not

describe properly the electronic structure of materials. The authors should present results obtained with a better quality functional (maybe the hybrid functional family). During the calculation, the surface is not specified which can influence the results. A schematic of the (2x2) and (4x4) supercells should be presented in SI, specifying the location of both vacancies and substituted atoms for the better understanding of the readers. Energetic values regarding formation of vacancies and substitution by anions are missing. Without such information, it is difficult to assess such process are favourable or not. The authors attribute the defect level to the 4d states of Mo, but actually in FigS1 it looks like S is also involved significantly. The positive/negative magnitude has to be explained elaborately. Authors explain that it is based on Bader charges, but for me it does not make sense that Mo has excess electrons (negative value). Furthermore, Bader charge values are extremely dependent on the functional used, since the electronic density can be not well described with PBE.

(2) The anomalous temperature dependence of the electronic transport of the material is interesting and in contrast to the previously reported 2D materials. The authors have not explained elaborately about the reason behind it. Is the activation energy playing any role? The authors may look for temperature dependent conductivity study or temperature dependent decay time of the photo-response study to get further insight regarding the role of traps and the activation energy on the carrier transport. It is very difficult to understand why at the certain value of critical temperature the thermal perturbation on de-trapping process weaken abruptly. The authors should clarify it with experimental facts.

(3) The interband defect spectra is important for photo-response study. The authors mentioned that the trap levels are in the range of 0.46 eV and 0.64 eV. However, there is a possibility that it can show photo-response with energy higher than 0.64 eV. The authors should perform some wavelength dependent quantum efficiency or responsivity measurement in the interband region to get the idea about trap state spectra in the bandgap. Otherwise, temperature dependent frequency variation of the capacitance can also be performed to find out the relative position and trap distribution inside the bandgap. The illumination and dark current ratio for 1550 nm and 1940 nm incident laser does not change with incident light intensity. Is it possible that no. of exited electrons are already more than the trapped hole in case of 1550 nm and 1940 nm excitation? Is it possible for authors to check the response with a much reduced light intensity on that particular wavelength?

(4) The infrared responsivity of the device is impressive. Can the authors comment on the detectivity of these devices?

(5) Figure 3d is not clear and well described in the text. Isn't it the response time for the real time device should be evaluated from fig. 4d where real-time program, read and erase is shown. It is not understandable to this reviewer. The authors should elaborate it.

(6) Is there any dependence of erasing speed for memory devices on the laser wavelength or is it wavelength and intensity independent?

(7) The manuscript shows impressive device performance for infrared photo-detection and opto-electronic non-volatile memory. Although the state of art with those device performances are missing from the text particularly for the photo-detectors. It is very important for the readers to assess the presented device performance with the available technology and device performance.

Reviewer #3 (Remarks to the Author):

This paper proposes utilization of inevitable defect states in TMDs as charge trapping centers for electronic memory applications and optical (IR) sensors. It is an interesting topic and application but the advantages of the TMDs for the specific applications are not clear since high mobility is not important in the relatively slow memory and sensing operations shown (e.g. ms for memory programming).

I believe the application is novel, but the conclusions are not well justified or supported by the experimental data provided, and hence the manuscript should not be published in its current

version.

The abstract indicates the electronic memory and IR sensor results were obtained for devices with individual nanosheets (this single layer of MoSSe?), but in the results section, the authors indicate that all data shown in the text is for a 7.6 nm sheet device.

The anomalous temperature dependence of the FET I_{ds} - V_{gs} curves (decreasing I_{off} from 300 K to 180 K but increasing I_{off} from 140K to 80K, with an abrupt change in behavior between 140 and 180 K) needs to be better understood and justified with experimental data, such as any leakage gate/substrate currents that may be present (not shown). The proposed mechanism may be a plausible explanation for the anomalous behavior but is not supported by experimental data.

It is not clear why retention would be characterized at high field ($V_g = -80$ V) and 80 K when retention usually refers to the stability of an undisturbed state, under equilibrium, normal conditions.

The proposed write/erase/read schemes are not clear, and why one electrical and one optical. The reason for these (electrical write and optical erase?) should be justified. The reason for reading at high gate field of -80 V is also not clear since reading is also typically done under low, non-disturbing gate and s/d fields.

The negative (only once) and typical positive photoresponse are confusing, and not consistent with the model proposed for the FET anomalous temperature dependence behavior, based on the minority carrier trapping and detrapping.

The use of deep states for minority carrier trapping for both memory and sensing operation is confusing and also not consistent with shown I_d - V_{gs} n-channel FET operation (which would require a p-type semiconductor channel to be inverted, in which holes would be the majority carriers, not the minority). Or holes would come from substrate under high positive gate field? Very minimal tunneling is expected through 280 nm oxide under 80 V (0.28 V/nm).

Very little detail is provided on the fabrication and structure of the devices. What are the dimensions, L, W of the FETs, what type and level doping is the Si substrate which acts as the back-gate of the FETs, what type of oxide is on the substrate, thermally grown or deposited? What are the intrinsic electronic properties of the semiconductor material (TMD flake), conductivity, mobility, carrier type and carrier concentration? These are important to interpret the I_d - V_{gs} curves provided.

The article is, in general, well written and organized but requires some English editing throughout. For example, the abstract starts with 'Defect plays' instead of 'defects play' which would be the common usage in the field.

The data plots are generally too small and difficult to read.

Response to Reviewer #1

We are grateful to the reviewer for the positive evaluation on the manuscript. In the revised manuscript, we have addressed the reviewer's comments and revised our manuscript accordingly. Our responses are listed below.

In the manuscript by Lei Yin et. al. titled "Robust trap effect in transition metal dichalcogenides for advanced multifunctional devices" its authors report the realization of robust defect-induced trap effect to construct the multifunctional electronic and optoelectronic devices. The authors discuss both the physical model and the electrical transport properties of the device in detail. The authors then show that the device shows infrared detection with very high photoresponsivity and high photoswitching ratio. Finally, they use the device to demonstrate the nonvolatile memory properties with high program/erase ratio and fast optical erasing time. The results are fairly impressive and are a fit for this journal. I recommend it be accepted after addressing the following issues:

Comment 1: *In Fig. 2, there are too many description about the temperature dependent properties of the device. However, these properties seems to have little relevance to the article. Could the authors add some explanation about the relationship between the temperature dependent properties and the multifunctional properties of the device.*

Our reply: We thank the reviewer for pointing out this issue. As discussed in the “**Anomalous temperature dependence of electronic transport**” section, our devices exhibit an anomalous temperature-dependent transport behavior: that is, the gate tunability weakened when the temperature dropped below a certain value (T_{critical}), causing a negligible current on/off ratio. This behavior is in contrast to the previously reported 2D FETs (*Nat. Commun.* 4, 2642, 2013; *Science* 349, 625, 2015), in which on/off ratio remains almost constant or increases with the decreasing temperature.

Actually, this anomalous temperature dependence is strongly relevant to the robust defect-induced trap effect proposed here. We would like to mention two important points here for the sake of adequately explaining the relationship between the temperature dependent properties and the multifunctional properties of the device.

1. The robust defect-induced trap effect, which is crucial to realize multifunctional properties (infrared detection and nonvolatile memory) of the device, is closely related to the temperature. The detailed study of temperature-dependent properties is very helpful to reveal the underlying mechanisms. In this research, supported by DFT calculations and electronic/optoelectronic characterizations, we demonstrated that such a robust trap effect is based on the stable capture of holes. As shown in Fig. 2b and 2c (see **Fig. R1** below), the charge shielding effect (e.g. anomalous high-current state at $V_{GS} = -80$ V) starts to appear at 140 K and becomes stable at 80 K, in accordance with the thermally-assisted holes' de-trapping process. This point has been discussed in the Manuscript.

Figure R1 | (a) I_{DS} - V_{GS} curves of device #1 under various temperatures. $V_{DS} = 1$ V. Inset, plot of the current on/off ratios along with temperatures. The corresponding critical temperature ($T_{critical}$), at which the on/off ratio starts to drop, is indicated by the red circle. (b) Time-resolved current measurement at $V_{GS} = -80$ V and $V_{DS} = 1$ V.

2. The study of temperature-dependent properties can help us to determine the optimum operating temperature of the devices, which is of vital importance to their

practical application. Thus, the detailed study of temperature-dependent properties is important and necessary.

Based on your comment, we have added the illustration to the revised Manuscript as “In this case, the holes’ de-trapping process starts to weaken at 140 K (defined as the critical temperature, T_{critical}), and is completely suppressed at 80 K. These results not only further solidify the above discussed physical model, but also help determine the optimum operating temperature of the device.” Please see the marked text on page 9, lines 18-22 in our revised Manuscript.

***Comment 2:** In the “Infrared memory and its operational mechanism” section, the authors use the electrical pulse to programme the device, and use the laser pulse to erase device. However, in normal memory device, people always use one kind of physical way to induce both programme and erase behavior. Could the authors demonstrate that your device can be programmed/erased only using the electrical pulse or laser pulse.*

Our reply: The operating mode of a memory device is determined by its component materials and device configuration. For nonvolatile memory applications, a device should be bistable, that is, ON and OFF states should be achieved at the same gate voltage (*ACS Nano* 9, 8089, 2015). Conventional flash memories are based on the all-voltage driven trapping/de-trapping of the charge carriers, corresponding to the erase and program (ON and OFF) states. However, the growing amount of data creates an urgent need for new and improved data storage methods. Benefiting from its high capacity, low energy consumption and long-distance secure communications, optical data storage has been considered as a highly promising alternative to conventional approaches (*Nat. Rev. Mater.* 1, 16070, 2016; *Science* 357, 1392, 2017). Specifically, the infrared radiations of 850, 1310 and 1550 nm with low energy dissipation in optical fibers are typical optical communication wavebands. Therefore, the optoelectronic devices that would convert and store infrared information into electrical signals, thereby enabling optical data communications, are highly pursued

(*Sci. Adv.* 4, eaap7916, 2018).

In this regard, we designed a nonvolatile infrared memory device. Its bistable behavior is based on the stable trapping of holes and their release. As for its all-voltage switching, as illustrated in **Fig. R2**, no obvious hysteresis (i.e., memory window) is observed in our device along the two different sweeping directions of gate voltage (V_{GS}), suggesting that only using the electrical pulse cannot define bistable states. Thus, a specific operating mode (electrical programming and optical erasing) is necessary for our infrared memory, just like all other reported 2D optoelectronic memory devices (*Nat. Commun.* 9, 2966, 2018; *Nat. Commun.* 8, 14734, 2017; *Sci. Adv.* 4, eaap7916, 2018; *Adv. Mater.* 31, 1807075, 2019; all of them must be electrically erased or programmed; 2D optoelectronic memory with all-optical switching has not yet been realized). And compared to these reported optoelectronic memories, our devices own simpler device configuration (just based on an individual TMD flake) and better performance. These have been detailedly discussed in the “**Infrared memory and its operational mechanism**” section of our Manuscript.

Figure R2 | I_{DS} - V_{GS} curve of the device along two different V_{GS} sweeping directions under dark state. No obvious hysteresis (i.e., memory window) is observed.

Comment 3: *Can the authors draw some schematic diagrams to explain the mechanisms of the device during the gate voltage-dependent memory operation?*

Our reply: Many thanks for this kindly reminder. To better explain the gate voltage

(V_{GS})-dependent memory effect, the schematic diagrams of various program processes and the corresponding read processes are redrawn and displayed in **Fig. R3**. As discussed in the Manuscript, holes can be stored into trap levels and retained after applying an appropriate V_{GS} pulse to the device. These trapped holes can act as charge shielding centers and provide sufficient shielding effect to the gate field of -80 V, thereby suppressing the depletion of electrons in the channel. Thus, a retentive low resistance state (LRS) can be read out at $V_{GS} = -80$ V. As can be seen from the newly introduced figures in the revised manuscript (Fig. 4e and 4f), by applying different programming V_{GS} (V_{pro}), we can get different readout states.

When V_{pro} is set in the range of -80 to -30 V (**Fig. R3a**), the strong electrostatic attraction between the large negative gate voltage and positive charges is not conducive to capture and store holes (**i**). Thus, in the absence of charge shielding centers, the electrons in the channel are depleted by large negative gate voltage, and the LRS (measured at $V_{GS} = -80$ V, $V_{DS} = 1$ V) cannot be effectively identified (**ii**).

Along with the increase of V_{pro} (-30 V $< V_{pro} < 0$ V, **Fig. R3b**), the LRS can be easily read out and gradually becomes stable. It suggests that holes start to be captured by defect levels (**iii**), which can provide stronger shielding effect to the gate field of -80 V (**iv**). Hence, $V_{pro} = -30$ V is a threshold voltage (V_T) for storing charges here.

Further, when a nonnegative V_{pro} is applied (**Fig. R3c**), the corresponding electrostatic attraction disappears, and the capture of holes becomes much easier (**v**, **vi**). Sufficient numbers of trapped holes give rise to a robust charge shielding effect (**vii**). Thus, very stable LRS can be obtained at $V_{GS} = -80$ V.

According to your suggestion, the redrawn schematic diagrams have been displayed in Fig. S27 of the revised Supplementary Information. The corresponding descriptions have also been revised. Please see the marked text on page 19 (lines 4-18) in our revised Manuscript.

Figure R3 | The schematic diagrams of the programming processes (i, iii, v and vi) under various programming V_{GS} (V_{pro}) and the corresponding LRS-read processes (ii, iv and vii) at $V_{\text{GS}} = -80 \text{ V}$. (a) $-80 \text{ V} < V_{\text{pro}} < V_{\text{T}}$; (b) $V_{\text{T}} < V_{\text{pro}} < 0 \text{ V}$; (c) $V_{\text{pro}} \geq 0 \text{ V}$. Electrons and holes are represented by yellow and red balls, respectively. Gray balls are the disappeared electrons and holes. Black dashed and solid lines in the bandgap denote the empty state (with trapped holes) and occupied state (without trapped holes), respectively. Black arrows in the bandgap represent the trapping holes.

Comment 4: *Further discussion needs to be added for power consumption of these devices during the gate voltage-dependent memory operation.*

Our reply: The reviewer raised a valid point. Our devices are operated via electricity

(for programming) and laser (for erasing) actuation. **Table R1** summarizes the corresponding power consumption during these two processes. And they are in the ranges of picojoule and femtojoule, respectively. These values are comparable to, even lower than that of other nonvolatile infrared memories (*Sci. Adv.* 4, eaap7916, 2018). Based on your comments, the above discussions have been added to page 18 (lines 6-10) of the revised Manuscript and Table S2 (page 31) in the revised Supplementary Information.

Table R1 | Power consumption of the infrared memory based on $\text{MoS}_{2x}\text{Se}_{2(1-x)}$.

	Device #1		Device #2	
	1550 nm	1940 nm	1550 nm	1940 nm
Power consumption	3.5 pJ/	3.5 pJ/	1.6 pJ/	1.6 pJ/
(programming/erasing)	8.4 fJ	25 fJ	8.8 fJ	27 fJ

Comment 5: *For tangible operation, the device should work at the level of tens of ms. Please compare the operation speed with previous reports. Do you have suggestion to improve the operation speed?*

Our reply: Both the electrical programming and optical erasing processes of our memory device have shown fast operation speed (0.3 ms and 0.9 ms, respectively). These values are superior to the previously reported 2D optoelectronic memories. Examples include WSe_2/BN heterostructures with optical erasing speed of 2 s (*Nat. Commun.* 9, 2966, 2018), MoS_2/PbS heterostructures with electrical erasing/optical programming speed of 0.17 s/15.7 ms (*Sci. Adv.* 4, eaap7916, 2018), and $\text{MoS}_2/\text{carbon nanotubes}$ heterostructures with electrical erasing/optical programming speed of 0.4 ms/32 ms (*Small* 15, 1804661, 2019). For ease of comparison, a table containing the figures of merit for different optoelectronic memories is incorporated. For detail, please refer Table S2 (page 31) in our revised Supplementary Information.

Such fast operation speed may be attributed to the simple device configuration. Unlike the heterostructure-based nonvolatile optoelectronic memories (as mentioned above), our devices are just based on an individual TMD flake. Thus, in the programming and erasing processes, there's no need for charge carriers to transfer across the hetero-interface. To further improve the operation speed of our devices,

adopting optimized contact metal or high- k dielectric materials could be an effective strategy, as it can speed up the carrier transport. Based on your comments, we have added the above discussions to page 18 (lines 8-14) of the revised Manuscript.

Comment 6: *There are some reports about the nonvolatile operation of 2-d materials based device? What is the difference between your device and the other devices reported by other groups.*

Our reply: As silicon-based flash memories are approaching their fundamental limit, novel materials and innovative device concepts are being investigated. Many 2D materials-based nonvolatile memories have been fabricated and shown superior device performance (*Nat. Commun.* 7, 12725, 2016; *Nat. Electron.* 1, 356, 2018). Recently, benefiting from its high capacity and long-distance secure communications, optoelectronic memory (especially infrared memory) has become one of the hottest topics. These have been detailedly illustrated in our **Reply 2**. In the currently reported 2D materials-based optoelectronic memories, the additional photosensitive or charge transfer/storage medium is indispensable. Examples include hetero-stacks (*Sci. Adv.* 4, eaap7916, 2018; *Nat. Nanotech.* 8, 826, 2013), floating-gated structure (*Nat. Nanotech.* 12, 901, 2017; *Adv. Mater.* 31, 1807075, 2019), and functional substrates (*Nat. Commun.* 9, 2966, 2018; *Nat. Commun.* 8, 14734, 2017).

Compared to these reported optoelectronic memories, our devices own simpler device configuration (just based on an individual TMD flake) and exhibit better performance. Its programmable and nonvolatile properties are realized by a robust defect-induced trap effect in TMDs. Supported by density functional theory calculations and device characterizations, we demonstrate experimentally that such a robust trap effect is introduced via the synergistic effect of S vacancy and substitutional Se atom. Although our device configuration is very simple, it exhibits superior memory performance, such as long-term stability, high program/erase ratio ($\sim 10^8$) and fast electrical programming/optical erasing time (0.3 ms/0.9 ms), providing a new technical direction in simplifying device fabrication procedures and

improving integration level. Based on your comments, a table containing more comparison information of different optoelectronic memories is given as Table S2 (page 31) in our revised Supplementary Information.

Comment 7: *In my opinion, your device with the nonvolatile memory and the photoelectricity properties is not novelty. However, the application of using these properties to demonstrated some new application would be interesting and novel. For example, these properties can be used to demonstrate some neuromorphic properties (memristor or synaptic emulation). Could the author give some discussions or prospects about the new application of the device?*

Our reply: Due to their great potential application in future electronic and optoelectronic devices, the nonvolatile memory and photoelectricity properties have gained increasing attention in both fundamental and applied science. One marked difference from conventional nonvolatile memory is that our optoelectronic memories are based on a robust defect-induced trap effect, introduced via the synergistic effect of vacancies and substitutional atoms. As a result, the whole process of nonvolatile memory and infrared detection can be completed in just one individual TMD flake, providing a new technical direction in simplifying fabrication procedures of the multifunctional device and improving integration level. In addition, because there is no need for charge carriers to transfer across the hetero-interface, our devices exhibit ultrafast operation speed. In view of the above, we believe that this device properties are sufficiently distinguishable from the previous reports.

We agree that neuromorphic electronics is one of the hot areas of research in recent years. In particular, synapses are the basic unit of the human brain for computing and memory. Consequently, using novel artificial devices to simulate synaptic short-term plasticity and long-term plasticity behavior is of great significance for the evolution of neuromorphic computation (*Science* 334, 623, 2011; *Adv. Mater.* 28, 3557, 2016.). As demonstrated in the “**Infrared memory and its operational mechanism**” section, the readout states are very sensitive to V_{pro} pulse applied to the device. Therefore, the

V_{pro} pulse and readout current can be regarded as the input spike and post-synaptic current, respectively. When V_{pro} pulse enhances gradually, the relaxation time of corresponding currents prolongs, implying its potential in emulating a long-term plasticity of synapsis.

Based on your comment, the above discussions have been added into the “**Infrared memory and its operational mechanism**” section, to illustrate the possible applications of our devices in neuromorphic electronics. Please see the marked text on page 19, lines 1-2 in our revised Manuscript.

Comment 8: *There are several grammatical and typos which make reading difficult. They need to be dealt with before this article is published.*

Our reply: Thanks for your suggestion. The whole manuscript and supplementary material have been revised by a native English speaker. Now we believe it has got much improved.

Response to Reviewer #2

We thank the reviewer for the instructive comments that stimulate us to further improve our manuscript. In the revised edition, we have addressed these comments and revised our manuscript accordingly. Our point-by-point responses are stated as below.

The manuscript by Yin et al. represents a report on advanced multifunctional devices based on metal dichalcogenides using defect/substitution induced traps. The authors explore a new way to use defects in their advantage contrasting to most of the device aspects where defects are attributed to the detrimental of the device performance. They have shown that using stable hole traps, both infrared photodetector as well as nonvolatile memory with reasonably high performance can be obtained. The detailed and elaborative device study along with DFT prediction for the trap formation is shown.

The device performance and the optoelectronic engineering of metal dichalcogenides sheet seems novel and interesting. However, several physical aspects of the device performance as well as the results claimed in the manuscripts are missing or unclear. In this present form, the manuscript cannot be published for a journal like Nature Communications. However, if the authors can address the following concerns and incorporate them in the manuscript, it can be considered for publication.

Comment 1: *The authors discussed DFT results obtained with PBE functional, which is well known to not describe properly the electronic structure of materials. The authors should present results obtained with a better quality functional (maybe the hybrid functional family). During the calculation, the surface is not specified which can influence the results. A schematic of the (2x2) and (4x4) supercells should be presented in SI, specifying the location of both vacancies and substituted atoms for the better understanding of the readers. Energetic values regarding formation of*

vacancies and substitution by anions are missing. Without such information, it is difficult to assess such process are favourable or not. The authors attribute the defect level to the 4d states of Mo, but actually in FigS1 it looks like S is also involved significantly. The positive/negative magnitude has to be explained elaborately. Authors explain that it is based on Bader charges, but for me it does not make sense that Mo has excess electrons (negative value). Furthermore, Bader charge values are extremely dependent on the functional used, since the electronic density can be not well described with PBE.

Our reply: We would like to thank the reviewer for the thoughtful scrutiny of our Manuscript. For a better illustration, we will response the reviewer's concerns point by point.

1. The authors discussed DFT results obtained with PBE functional, which is well known to not describe properly the electronic structure of materials. The authors should present results obtained with a better quality functional (maybe the hybrid functional family).

We agree that compared to hybrid functional (HSE06), PBE functional is indeed at a disadvantage in describing the bandgap of materials. And according to the reviewer's comments, the band structures of (2×2) supercell of MoS_2+SV and $\text{MoS}_{2x}\text{Se}_{2(1-x)}+\text{SV}$ calculated by HSE06 have been added to Fig. S4 in the revised Supplementary Information (see **Fig. R4** below). However, we would like to mention three important points here for the sake of adequately explaining why used PBE functional is.

(a) In our research, in order to modify the properties of these defect levels caused by anion vacancies in TMDs, we need to understand their underlying origin. Thus, as mentioned in our Manuscript, the important conclusion that "deep defect levels can be induced by SV and mainly attributed to the 4d states of Mo atom around the SV" is obtained by DFT calculations at PBE level. The result is consistent with the previous reports (*Nat. Commun.* 4, 2642, 2013; *ACS Nano* 12, 2569, 2018), where PBE functional was also used. The above discussions have been added to page 20 (lines

23-26) in the revised Manuscript.

(b) Compared to hybrid functional, PBE functional only underestimates the bandgap value of TMDs, but the overall shape does not change significantly. It has been certified by previous reports (*J. Mater. Chem. A* 2, 7960, 2014). For our case, the band structures of (2×2) supercell of MoS_2+SV ($\text{MoS}_{2x}\text{Se}_{2(1-x)}+\text{SV}$) obtained from HSE06 and PBE functional own the same overall shape (**Fig. R4**). Moreover, the deep defect levels can still be induced by SV and mainly attributed to the 4d states of Mo atom around the SV, according to the calculations at HSE06 level.

(c) Actually, it is difficult to calculate the band structure of a (4×4) supercell (containing 47 atoms) at HSE06 level. Moreover, our research focuses on the properties of defect levels caused by SV in TMDs, such as their underlying origin and the regulation of selenium atoms on them. These results will be not affected by the functional used.

Figure R4 | Comparison between the band structures of (2×2) MoS_2 and $\text{MoS}_{2x}\text{Se}_{2(1-x)}$ monolayer with one SV at HSE06 level (a) and PBE level (b).

2. During the calculation, the surface is not specified which can influence the results. A schematic of the (2×2) and (4×4) supercells should be presented in SI, specifying the location of both vacancies and substituted atoms for the better understanding of the readers.

To better understand the computational model, we supplemented the schematic of the (2×2) and (4×4) supercells in Fig. S2 and S5 of the revised Supplementary Information (see **Fig. R5** and **R6** below), respectively. Both the locations of S vacancy (highlighted by pale yellow circle with black outline) and substituted atoms (highlighted by black ball) are on the upper surface of MoS_2 . The corresponding descriptions have also been added to page 20 (lines 17, 19-20) in the revised Manuscript.

Figure R5 | Atomic structure of (2×2) supercell of monolayer MoS_2 and $\text{MoS}_{2x}\text{X}_{2(1-x)}$ ($X = \text{O}, \text{Se}, \text{Te}$) without (a-d) and with (e-h) one SV. Both top views (left) and side views (right) are shown. The bond length of Mo-X ($X = \text{S}, \text{O}, \text{Se}$ or Te) is labeled on the bottom of side view. Mo, S, O, Se and Te atoms are represented by blue, yellow,

red, black and orange balls respectively. The dotted line on the side view of atomic structure is used to show the structural change of $\text{MoS}_{2x}\text{X}_{2(1-x)}$ clearly. The SV is highlighted by pale yellow circle with black outline.

Figure R6 | (a, b) Atomic structure of (4×4) supercell of monolayer MoS_2 without (a) and with (b) one SV. (c, d) Atomic structure of (4×4) supercell of Se-doped MoS_2 with one SV, in which Se atom is close to (c) and far from (d) the SV. The SV is highlighted by pale yellow circle with black outline. Mo, S and Se atoms are represented by blue, yellow and black balls, respectively. The formation energy of one SV in pristine MoS_2 and Se-doped MoS_2 is labeled on the bottom of (b), (c) and (d), respectively.

3. Energetic values regarding formation of vacancies and substitution by anions are missing. Without such information, it is difficult to assess such process are favourable or not.

The formation energy of vacancy (E^{form}) and substitution by anions (E^{sub}) are given in **Fig. R6** and **R7**, respectively. The formulas for calculating E^{form} and E^{sub} are as follows (*Phys. Chem. Chem. Phys.* 20, 16077, 2018),

$$E^{\text{form}} = E(\text{defect}) - E(\text{perfect}) + \mu_{\text{S}},$$

$$E^{\text{sub}} = E(\text{Se+slab}) - E(\text{slab}) + \mu_{\text{S}} - \mu_{\text{Se}},$$

where $E(\text{defect})$ and $E(\text{perfect})$ represent respectively the total energy of a (4×4) supercell with and without one SV, $E(\text{Se+slab})$ and $E(\text{slab})$ represent respectively the total energy of a (4×4) supercell with and without one Se atom, μ_{S} and μ_{Se} are the chemical potential of a S and Se atom. The computed values of μ_{S} and μ_{Se} chemical potentials are -3.464 and -3.309 eV/atom, which are obtained from bulk (space group: $\overline{\text{R3mH}}$) S and Se, respectively. Here, we focus on the E^{form} (E^{sub}) differences of an SV (substituted Se atom) at different sites, which is independent of the choice of μ_{S} (μ_{Se}) (*Phys. Rev. Lett.* 115, 126806, 2015).

Firstly, we discussed the formation energy of SV. There are three sites of SV in pristine MoS_2 and Se-doped MoS_2 , as illustrated in **Fig. R6b-d**. Their formation energies are 3.04, 3.01 and 3.01 eV, respectively. A positive SV formation energy means that the formation of SV is an endothermic reaction. The smaller the formation energy, the more easily it can form the SV. Therefore, SV is easier to form in Se-doped MoS_2 , and its formation location is independent of the location of the substituted Se atom.

Secondly, we discussed the formation energy of substitution by Se atom in the presence of an SV. As shown in **Fig. R7**, there are three locations for Se atom (labeled by the number of 1, 2 and 3) in the (4×4) supercell. From location 1 to location 3, the distance between the substituted Se atom and the SV increases gradually. The formation energy of substitution of the nearest-neighbor site (location 1) and the next-nearest-neighbor site (location 2) is equal (0.828 eV), while the formation energy of substitution of location 3 is the largest (0.834 eV). It means that Se atom is more favorable to substitute S atom near SV. Therefore, when we model Se concentration (C_{Se}) from 3.13% (one Se atom) to 25.00% (eight Se atoms) in Se-doped MoS_2 with one SV, the S atoms at location 1 are prior to be replaced by Se atoms, then followed by location 2 and location 3 (**Fig. R8**).

Based on your comment, the above discussions have been added to the revised

Manuscript and Supplementary Information. For details, please see the marked text on page 6 (lines 13-17) in our revised Manuscript and Fig. S5-S7 in the revised Supplementary Information.

Figure R7 | Schematic diagram of all possible substitution locations of one Se atom in a (4×4) supercell of MoS₂ with one SV. Blue balls represent the cation sites. And other colorful (yellow, red, dark blue and pink) balls represent the anion sites. Location 1, location 2 and location 3 are shown by red, dark blue and pink balls, respectively. The SV is highlighted by pale yellow circle with black outline. From location 1 to location 3, the distance between the substituted Se atom and the SV increases gradually. The formation energy of substitution at location 1, 2 and 3 is labeled on the right.

Figure R8 | The atomic structure of $\text{MoS}_{2x}\text{Se}_{2(1-x)}$ with increasing C_{Se} from 3.13% to 25.00%. Blue, yellow and black balls represent the Mo, S and Se atoms, respectively. The SV is highlighted by pale yellow circle with black outline.

4. The authors attribute the defect level to the 4d states of Mo, but actually in FigS1 it looks like S is also involved significantly.

As shown in Fig. S1 of Supplementary Information, the larger the plotted point is, the stronger the corresponding electron states contribute to the defect levels. It is true that S-3p states participate in the contribution of defect level. However, its contribution is weaker than that of Mo-4d states. When the regulation of Se atoms on defect levels is considered, Mo-4d states are more likely to be affected compared with S-3p states. Hence, we mainly focus on the change in contribution of Mo-4d states to the defect levels after introducing Se atoms. Based on your comment, we have revised the corresponding description as “The deep defect levels can be mainly attributed to the 4d states of Mo atom around the SV”. Please see the marked text on page 5 (line 2)

in the revised Manuscript, and the caption of Fig. S1 in the Supplementary Information.

5. The positive/negative magnitude has to be explained elaborately. Authors explain that it is based on Bader charges, but for me it does not make sense that Mo has excess electrons (negative value). Furthermore, Bader charge values are extremely dependent on the functional used, since the electronic density can be not well described with PBE.

Note that no partitioning scheme is available to rigorously determine the atomic charges because the concept of atomic charges is actually an approximation. Bader charge is usually applied to monitor the atomic charges state (*Science* 329, 1633, 2010; *Comput. Theor. Chem.* 976, 153, 2011) and analyze the electron transfer in materials (*Phys. Rev. B* 90, 134102, 2014). Hence, we utilize Bader charge as a reference to monitor the charge state of the Mo atoms around SV in MoS₂ and Se-doped MoS₂, which can further indicate the ability to lose electrons of Mo atoms around SV. As mentioned in our Manuscript, the magnitude of negative/positive value denotes the ability to lose/obtain electrons, i.e., the Bader charge value from negative to positive means that the ability to lose (obtain) electrons is gradually weakened (enhanced).

Taking MoS₂+SV model as an example. There are 6 valence electrons (4d⁵s¹) in an isolated Mo atom. While the valence electron of Mo1 atom in MoS₂+SV (**Fig. R9**) becomes 5.35 after performing the Bader charge calculation. It indicates that Mo1 atom **loses** 0.65 |e| in the MoS₂+SV at PBE level, thereby defining the Bader charge value as **-0.65 |e|**. Here, negative value represents loss of electrons.

Based on your comments, the detailed explanations for positive/negative magnitude have been added to the Manuscript as “where positive/negative sign denotes the acquisition/loss of electrons, and the Bader charge value from negative to positive means that the ability to lose (obtain) electrons is gradually weakened (enhanced)”. Please see the marked text on page 5, lines 17-19 in our revised Manuscript.

Additionally, we also calculated the Bader charge value using HSE06 functional.

As shown in **Fig. R9**, it is true that the obtained values with PBE and HSE06 functional are different. However, the variation tendency of charge value (toward the positive direction) after the introduction of Se atom is consistent. Therefore, the important conclusion that “the ability to lose electrons of Mo atoms around SV is weakened by substituting S with Se atom, and the electronegativity of Mo atoms is enhanced” will not be affected. Actually, Bader charge analysis have also been successfully described by PBE functional (*Phys. Rev. B* 98, 014109, 2018; *Phys. Rev. Lett.* 121, 157001, 2018). The corresponding descriptions and Fig. R9 have been added to page 20 (line 22-23, 25-26) in the revised manuscript and Fig. S4 in the revised Supplementary Information.

Mo location	PBE level		HSE06 level	
	MoS ₂ +SV	MoS _{2x} Se _{2(1-x)} +SV	MoS ₂ +SV	MoS _{2x} Se _{2(1-x)} +SV
Mo1	-0.65 e	+0.24 e	-1.27 e	-0.82 e
Mo2	-0.65 e	-0.19 e	-1.42 e	-1.25 e
Mo3	-0.68 e	+0.94 e	-0.80 e	-0.10 e

Figure R9 | The Bader charge of (4 × 4) MoS₂ and MoS_{2x}Se_{2(1-x)} monolayer with one SV at PBE and HSE06 level. The SV is highlighted by black dotted circle. Mo, S and Se atoms are represented by blue, yellow and black balls, respectively. The location of selected Mo atoms is marked on top panel and the corresponding Bader charge values are listed at bottom panel.

Comment 2: *The anomalous temperature dependence of the electronic transport of the material is interesting and in contrast to the previously reported 2D materials. The authors have not explained elaborately about the reason behind it. Is the activation*

energy playing any role? The authors may look for temperature dependent conductivity study or temperature dependent decay time of the photo-response study to get further insight regarding the role of traps and the activation energy on the carrier transport. It is very difficult to understand why at the certain value of critical temperature the thermal perturbation on de-trapping process weaken abruptly. The authors should clarify it with experimental facts.

Our reply: Thanks for bringing this issue to our attention. According to this insightful comment, temperature-dependent conductivity has been studied in detail, as discussed below. For device #2 with a good ohmic contact under 300 K and 80 K (**Fig. R10a,b**), the two-terminal conductivity is defined as $\sigma = I_{DS}/V_{DS} \times L/W$ (*ACS Nano* 11, 7362, 2017) and displayed as a function of temperature under various V_{GS} (**Fig. R10c**). The charge transport is thermally activated at $160 \text{ K} < T < 300 \text{ K}$ based on the good agreement of the data with the activation transport model: $G(T) = G_0 \exp(-E_a/k_B T)$, where k_B is the Boltzmann constant, E_a is the activation energy, and G_0 is the fitting parameter (*Nat. Mater.* 12, 815, 2013; *Nano Lett.* 16, 6383, 2016). The activation energy, specifying the energy difference between the Fermi level and mobility edge, is extracted and analyzed with the thermally activated transport. As shown in **Fig. R10d**, the activation energy is dependent on gate voltage, and is lower than that of other 2D materials on SiO_2 substrate (*Nano Lett.* 16, 6383, 2016; *ACS Nano* 8, 4859, 2014). The value of E_a is $\sim 1 \text{ meV}$ at $V_{GS} = -40 \text{ V}$. Such low E_a value verifies the proposed model of holes (minority carriers) trapping and de-trapping for explaining the anomalous electronic behavior.

Figure R10 | (a, b) I_{DS} - V_{DS} curves of device #2 under various V_{GS} from -30 V to -70 V. $T = 300$ K and 80 K. (c) The temperature and gate voltage dependence of conductivity curves of the device. (d) E_a curve as a function of V_{GS} .

Because the Fermi level (E_f) is very close to the mobility edge or the conduction band minimum (*ACS Nano*, 5, 7707, 2011), the unoccupied defect levels upon E_f are very shallow, and difficult to trap electrons stably even under low temperature (**Fig. R11a**). By contrast, the deep defect levels below E_f can trap and store holes more stably at low temperature, due to the weak thermal perturbation. These trapped holes with positive charge can effectively screen the effect of negative gate voltage (V_{GS}) and suppress the depletion of electrons in the channel, generating the anomalous high-current state at negative V_{GS} (**Fig. R11b**). This model is further supported by (a) the photoresponse measurement with infrared lasers (1550, 1940 and 2699 nm) and (b)

time-resolved current measurement at various temperatures (at $V_{GS} = -80$ V), which demonstrate the existence of in-gap traps and the influence of temperature on de-trapping process, respectively. For details regarding the photoresponse measurement, please see our **Reply 3**. The influence of temperature on de-trapping process is elaborated below.

Figure R11 | (a) Schematic band diagrams of a typical n-type semiconductor with various defect levels. Excess minority carriers (holes) will be captured and stored in *deep traps* under non-equilibrium conditions. Electrons and holes are represented by blue and red circles with e^- and h^+ , respectively. (b) Schematic band diagrams of $\text{MoS}_{2x}\text{Se}_{2(1-x)}$ devices at $V_{GS} < 0$ V, $V_{DS} > 0$ V. Electrons and holes are represented by yellow and red balls, respectively. Black dashed lines in the bandgap denote the empty state (with trapped holes).

The effect of thermal perturbation on holes' de-trapping process is characterized by de-trapping time (τ_t): $\tau_t^{-1} = s_p N_v v_{th} \exp(-\Delta E/(kT))$, where s_p , N_v , v_{th} , ΔE , and k are the capture cross section of the trap center, the effective density of states in the valence bands, the thermal velocity of the carriers, the energy difference between the trap state and the valence band edge, and the Boltzmann constant, respectively. At a high temperature, the provided thermal energy (kT) can promote the de-trapping process by thermally exciting the trapped holes back into the valence band. This rapid process makes it difficult to observe the effects of trapping holes. With the decrease in temperature, it will be difficult to thermally excite the trapped holes back to the

valence band due to the low thermal energy. It suggests that the effect of thermal perturbation on holes' de-trapping process starts to weaken. And the de-trapping time increases exponentially with the decrease of temperature. At this point, we can easily detect the anomalous electronic transport properties induced by the trapped holes. The time-resolved current measurement under various temperatures (at $V_{GS} = -80$ V) demonstrates the temperature-dependent holes' de-trapping process.

It should be noted that the effect of thermal perturbation on de-trapping process **does not weaken abruptly** (at the certain value of critical temperature). In our Manuscript, critical temperature ($T_{critical}$) refers to the temperature at which the effect of thermal perturbation on de-trapping process starts to weaken. In this scenario, a prolonged de-trapping time enables us to observe the anomalous temperature dependence of the electronic transport at $T_{critical}$ (140 K). However, the effect of thermal perturbation on de-trapping process is not completely removed at $T_{critical}$. As demonstrated in **Fig. 2c** (see **Fig. R12** below), the anomalous high-current state at $V_{GS} = -80$ V (140 K) decreases gradually as the test time increases, corresponding to the gradual de-trapping process. Further, when the temperature drops to 80 K, due to the lower thermal energy, the de-trapping process is completely repressed and the high-current state at -80 V is very stable. These results can help us to determine the optimum operating temperature of the device, which is of vital importance to their practical application.

Figure R12 | Time-resolved current measurement under various temperatures. $V_{GS} = -80$ V, $V_{DS} = 1$ V.

Based on your critical suggestion, more discussions about activation energy have been added into the “**Anomalous temperature dependence of electronic transport**” section of our revised Manuscript. Besides, the corresponding descriptions about the effect of thermal perturbation on holes’ de-trapping process have also been revised, to elucidate it more clearly. For details, please see the marked text on page 8-9 in our revised Manuscript and Fig. S12 and S13 in the revised Supplementary Information.

Comment 3: *The interband defect spectra is important for photo-response study. The authors mentioned that the trap levels are in the range of 0.46 eV and 0.64 eV. However, there is a possibility that it can show photo-response with energy higher than 0.64 eV. The authors should perform some wavelength dependent quantum efficiency or responsivity measurement in the interband region to get the idea about trap state spectra in the bandgap. Otherwise, temperature dependent frequency variation of the capacitance can also be performed to find out the relative position and trap distribution inside the bandgap. The illumination and dark current ratio for 1550 nm and 1940 nm incident laser does not change with incident light intensity. Is it possible that no. of exited electrons are already more than the trapped hole in case of 1550 nm and 1940 nm excitation? Is it possible for authors to check the response with a much reduced light intensity on that particular wavelength?*

Our reply: According to the wavelength-dependent photoresponse measurement with 473 nm (2.62 eV), 1550 nm (0.8 eV), 1940 nm (0.64 eV) and 2699 (0.46 eV) nm lasers, we verified the existence of trap levels and concluded that trap distribution should mainly concentrate in the range of 0.46 eV and 0.64 eV above the valence band edge (page 15, lines 2-7 in the Manuscript). This is determined by the unique photodetection mechanism of our devices. As mentioned in the Manuscript, the trapped holes with positive charge can effectively screen the negative gate voltage and suppress the depletion of electrons in the channel, generating a high dark current state at negative gate voltages. However, after the first illumination with 473, 1550 and

1940 nm lasers, the drain current reduced to the order of $\sim 10^{-13}$ A (i.e., the depletion of electrons in the conduction band), as shown in Fig. 3b (see **Fig. R13** below). This is attributed to the vanishing charge shielding effect, originating from the recombination between the trapped holes and photo-excited electrons (i.e., the trapped holes are erased by lasers).

Figure R13 | Time-resolved photoresponse measurement with 473, 1550 and 1940 nm lasers at $V_{GS} = -80$ V. The blue rectangle represents the state of illumination.

To further support the conclusion with evidence, we also conducted photoresponse measurement with the other two wavelengths: 808 nm (1.53 eV) and 3000 nm (0.41 eV) (**Fig. R14b, f**). The corresponding photoresponsivity are displayed in **Fig. R14g**. Obviously, there is no photoresponse under the illumination of 2699 and 3000 nm, suggesting that the low-energy photons (0.46 eV) are not sufficient to cause an interband light absorption. When the photon energy exceeds 0.64 eV, as the Reviewer mentioned, the apparent negative photoresponse can always be observed (**Fig. R14a-d**). Thus the photon energy of 0.64 eV is enough to cause an interband light absorption, and eliminate the charge shielding effect of trapped holes. We agree that few trap states may exist above 0.64 eV. But for our case, the trap levels are mainly in the range of 0.46 eV and 0.64 eV, which can be understood as follows. If most of traps are distributed at a position higher than 0.64 eV, the holes captured by these traps would not be erased under the illumination of 1940 nm, due to the insufficient photo-excitation energy. The corresponding charge shielding effect will be

maintained, and the electrons won't be depleted at large negative gate voltages. This is inconsistent with our experimental phenomena (**Fig. R14d**). In view of the above, we consider the main distribution of trap states is not higher than 0.64eV above the valence band edge.

We specifically thank you for pointing out this question, we have modified the corresponding text for more accurate description as “It's worth noting that when the photon energy exceeds 0.64 eV, the apparent negative photoresponse can always be observed (Supplementary Fig. 21). However, no obvious photoresponse was observed under 2699 nm (0.46 eV) and 3000 nm (0.41 eV) lasers (Supplementary Fig. 23). Therefore, we estimate that trap distribution should mainly concentrate in the range of 0.46 eV and 0.64 eV above the E_v ”. For details, please see the marked text on page 15, lines 2-7 in our revised Manuscript.

Figure R14 | (a-f) I_{DS} - V_{GS} curves of device #1 under dark and illuminated states with

473, 808, 1550, 1940, 2699 and 3000 nm lasers, $V_{DS} = 1$ V, $T = 80$ K. (g) The corresponding photoresponsivity (R_{ph}) as a function of power density at $V_{GS} = -80$ V.

As for the power density dependence of “the illumination and dark current ratio” (photo on/off ratio), within our experimental conditions, the ratio for 1550 nm and 1940 nm incident laser does not change with the power intensity (**Fig. 3e**). It is reasonable to anticipate that the number of excited electrons is already more than the trapped holes (i.e., the photoresponse reaches saturation), even performed with the lowest effective output power density of our lasers. We agree that when applying a very weak light signal, it is possible that the trapped holes won't be erased completely, thus generating a power density-dependent photo on/off ratio. However, we could not be able to conduct with a lower power density due to the limitation of instrument. Based on your comments, more discussions about the possibility with a lower power density have been added to page 14 (lines 8-14) of the revised Manuscript.

Comment 4: *The infrared responsivity of the device is impressive. Can the authors comment on the detectivity of these devices?*

Our reply: The specific detectivity (D^*) is indeed an important photodetectors figure of merit. It characterizes the capacity of detection for the lowest optical signal and is always described as $D^* = (AB)^{1/2}/NEP$, where A and B are the active area and bandwidth of photodetectors, respectively, and NEP (Noise Equivalent Power) is defined as the minimum incident radiation power demanded to realize an SNR (Signal to Noise Ratio) of 1 in a 1 Hz bandwidth. In order to obtain a true detectivity, noise measurements should be performed for an exact noise density and NEP (*Sci. Adv.* 3, e1700589, 2017; *Nat. Photonics* 12, 601, 2018). However, the professional test equipment is not accessible for us. Actually, D^* is generally estimated by using a simplified formula in most of research on photodetectors (*Science* 325, 1665, 2009; *Nano Lett.* 16, 2254, 2016; *Nat. Commun.* 8, 1906, 2017; *Adv. Mater.* 31, 1806725, 2019). Assuming shot noise from the dark current is the major contributor, specific

detectivity can be expressed as $D^* = I_{\text{ph}}A^{1/2}/PA(2qI_{\text{dark}})^{1/2}$, where I_{ph} ($I_{\text{ph}} = I_{\text{illumination}} - I_{\text{dark}}$) is the photocurrent, A is the active area of photodetectors, P is the laser power density, q is the elementary charge, and I_{dark} is the dark current. Note that this method tends to significantly overestimate the value of detectivity due to the neglect of other noise sources, i.e., Johnson noise and thermal fluctuation “flicker” noise (*Science* 325, 1665, 2009). Apart from its inaccuracy, this calculation method is often applied to the normal photodetectors with a positive photoresponse ($I_{\text{ph}} > 0$). Therefore, the method is not suitable to estimate the detectivity of our devices, which exhibits a negative photoresponse ($I_{\text{ph}} < 0$). As for such photodetector with a negative photoresponse, other parameters, such as photocurrent and gain, are generally used to quantify the photoresponse performance (*Nat. Nanotech.* 9 273, 2014; *Nano Lett.* 15, 5875, 2015). Thus, in addition to the photoresponsivity, we also calculated the photocurrent and gain of our devices. For detail please see Fig. S22 in the revised Supplementary Information.

Comment 5: *Figure 3d is not clear and well described in the text. Isn't it the response time for the real time device should be evaluated from fig. 4d where real-time program, read and erase is shown. It is not understandable to this reviewer. The authors should elaborate it.*

Our reply: We are sorry for the unclear descriptions of the response time. The response time shown in Fig. 3d was the test value for the real-time device, not the one extracted from Fig. 4d. This real transient response time was measured by an oscilloscope method (*Adv. Electron. Mater.* 2, 1600291, 2016; *Nat. Commun.* 10, 806, 2019; *Nat. Commun.* 5, 5404, 2014), where our device was connected in series with a 1 M Ω resistor. When the device is excited by external signals, an oscilloscope is used to record the bias change of the device, and to probe the current variation of the series circuit. In order to highlight the specific process of response, we only focused on the rising edge (falling edge) region triggered by the laser (voltage) pulse signal, which could lead to an unclear presentation.

According to the reviewer's comments, we provided a full cycle of response time measurement (**Fig. R15a**), and defined the response time in details. Firstly, the device is set to a high resistance state (HRS) at $V_{GS} = -80$ V. **(a)** When a gate voltage pulse of 80 V is applied, the holes are stored into trap levels and become charge screening centers (electrical programming process). Then, the device turns into and remains in a low resistance state (LRS) at $V_{GS} = -80$ V. During this process, due to the decreased resistance of the device, the current of the whole series circuit increases. As a result, the voltage drop across the load resistor increases, and the voltage drop across the device decreases. This falling time corresponds to the response time of electro-excitation (~ 0.3 ms), as shown in **Fig. R15b**. **(b)** When a laser pulse (1550 nm) is applied to the device, the stored holes in defect states are recombined by the photo-excited electrons (optical erasing process). As a result, the device returns to the original HRS, leading to the decreased current in the whole series circuit. Accordingly, the voltage drop across the device increases. The corresponding response time of photo-excitation is the rising time of 0.9 ms (**Fig. R15c**). From the partially enlarged view, a spike is observed at the moment when the laser is turned on, but its duration is very short. Such phenomenon has been widely observed in 2D material-based devices (*Nat. Nanotech.* 8, 826-830, 2013; *Adv. Mater.* 31, 1807075, 2019). Therefore, the response time is defined as the time interval between two steady states. Fig. R15 has been displayed in Fig. 3d of the revised Manuscript and Fig. S20 of Supplementary Information. Detailed descriptions about response time measurement have been added to page 13 (lines 19-22) and “**Methods - Device fabrication and characterization**” section of the revised Manuscript.

Figure R15 | (a) A full cycle of response time measured by an oscilloscope method. (b, c) Response time of the electro- and photo-excitation process. The laser wavelength is 1550 nm.

Comment 6: *Is there any dependence of erasing speed for memory devices on the laser wavelength or is it wavelength and intensity independent?*

Our reply: The erasing speed of memory with various laser wavelengths (473, 1559 and 1940 nm) is provided in **Fig. R16**. The data is collected by the oscilloscope method described in **Reply 5**. **Fig. R16d-f** shows the partially enlarged view of the rising edge region triggered by the laser signal, which was extracted from **Fig. R16a-c** to help us identify the response time. As for 1550 and 1940 nm lasers, the erasing time are almost the same (0.9 and 0.8 ms). This is because the low-energy photons of 1550 nm (0.8 eV) and 1940 nm (0.64 eV) lasers can only pump electrons to the position of

in-gap defect levels. Therefore, we consider the erasing time is independent of these infrared lasers. Noteworthy, when the same method is used to identify the erasing time triggered by a 473 nm laser pulse (**Fig. R16f**), the obtained time (0.2 ms) is faster than that of infrared illumination conditions. This is because the photoresponse with 473 nm laser is a result of the combined effect of interband and band-edge light absorption. The high-energy photons (2.62 eV) can excite electrons not only to the trap levels but also to the conduction band. As a result, the device does not return to original the HRS even if the stored holes have been erased. This is further verified by the fact that an additional rising edge occurs at the moment when the laser is turned off (**Fig. R16c**).

Based on your comments, the above discussions have been added to page 18 (lines 4-6) of the revised Manuscript and the caption of Fig. S26 in Supplementary Information.

Figure R16 | (a-c) The erasing speed measurement of the optoelectronic memory with various laser wavelengths (473, 1559 and 1940 nm). (d-e) The partial enlarged view of the rising edge region triggered by the laser signal.

Comment 7: *The manuscript shows impressive device performance for infrared photo-detection and opto-electronic non-volatile memory. Although the state of art*

with those device performances are missing from the text particularly for the photo-detectors. It is very important for the readers to assess the presented device performance with the available technology and device performance.

Our reply: Thanks for your kind suggestion, two new tables containing the figures of merit for different infrared photodetectors (**Table R2**) and nonvolatile optoelectronic memories (**Table R3**) are incorporated. These tables ease the comparison of our devices with other state-of-the-art devices reported in literatures. They have been added to our revised Supplementary Information as Table S1 and S2 (page 31).

Table R2 Comparison of the figures of merit for different infrared photodetectors based on 2D materials

Materials/structure	Measurement conditions	Photoresponsivity ($A W^{-1}$)	Response time	Ref.
MoS _{2x} Se _{2(1-x)}	$V_{GS} = -80 V, V_{DS} = 1 V, \lambda = 1940 \text{ nm}$	5.5×10^4	0.8 ms	This work
	$V_{GS} = -80 V, V_{DS} = 1 V, \lambda = 1550 \text{ nm}$	2.4×10^5	0.9 ms	
Black Phosphorus (BP)	$V_{GS} = 3 V, V_{DS} = 0.5 V, \lambda = 3.39 \mu\text{m}$	82		9
Bi ₂ O ₂ Se	$V_{DS} = 0.6 V, \lambda = 1200 \text{ nm}$	65	1 ps	10
PbS/MoS ₂	$V_{GS} = 20 V, V_{DS} = 2 V, \lambda = 800 \text{ nm}$	4.5×10^4	7.8 ms	11
MoS ₂ /HgTe quantum dot	$V_{GS} = -15 V, V_{DS} = 1 V, \lambda = 2 \mu\text{m}$	5×10^3		12
MoS ₂ /graphene/WSe ₂	$V_{GS} = 0 V, V_{DS} = 1 V, \lambda = 940 \text{ nm}$	0.306		13
h-BN/MoTe ₂ /graphene/ SnS ₂ /h-BN	$V_{GS} = 0 V, V_{DS} = 1 V, \lambda = 1550 \text{ nm}$	10	3.5 s	14

Table R3 Comparison of the figures of merit for different nonvolatile optoelectronic memories

Materials/structure	Program/erase ratio	Retention time	Programming/erasing time	Power consumption (programming/erasing)	Response wavelength	Ref.
MoS _{2x} Se _{2(1-x)}	10^8	10^4 s	0.3 ms/0.2 ms	3.5 pJ/49 fJ	473 nm	This work
			0.3 ms/0.9 ms	3.5 pJ/8.4 fJ	1550 nm	
			0.3 ms/0.8 ms	3.5 pJ/25 fJ	1940 nm	
CuIn ₇ Se ₁₁	<10	50 s			635 nm	15
MoS ₂ /functionalized substrates	4700	10^4 s			450 nm	16
PbS/MoS ₂	600	10^4 s	15.7 ms/0.17 s	420 pJ/1.5 pJ	808 nm	17
					1340 nm	
					1550 nm	

					1940 nm	
WSe ₂ /BN	10 ⁶	10 ⁴ s	2 s		473 nm	
					515 nm	18
					638 nm	
MoS ₂ /SWCNTs	10 ⁶	10 ³ s	32 ms/0.4 ms		532 nm	19

Response to Reviewer #3

We thank the reviewer for the instructive comments that stimulate us to further improve our manuscript. In the revised edition, we have addressed these comments and revised our manuscript accordingly. Our point-by-point responses are stated as below.

Reviewer's comment: *This paper proposes utilization of inevitable defect states in TMDs as charge trapping centers for electronic memory applications and optical (IR) sensors. It is an interesting topic and application but the advantages of the TMDs for the specific applications are not clear since high mobility is not important in the relatively slow memory and sensing operations shown (e.g. ms for memory programming).*

I believe the application is novel, but the conclusions are not well justified or supported by the experimental data provided, and hence the manuscript should not be published in its current version.

Our reply: We are happy to hear that the reviewer thinks our work is an interesting topic and the application is novel. 2D layered TMDs have gained increasing attention in both fundamental and applied science due to their unique structures and properties, such as diverse band structures, high flexibility and atomic-level flatness (*Science* 353(6298), aac9439, 2016; *Nat. Rev. Mater.* 2, 17033, 2017). Apart from its high mobility which can enable high-speed operation for transistors, the feature we mostly focus on is their high susceptibility to intrinsic defects. It originates from their atomic-scale thickness and high surface-to-volume ratio and has typically been considered to be a limiting factor in device performance.

In this work, taking advantage of this high susceptibility to intrinsic defects, we demonstrated experimentally that a functional defect (which can capture carriers and store them steadily) can be introduced in an individual TMD flake and be applied to high-performance infrared photodetector and nonvolatile optoelectronic memory,

providing a new technical direction in designing electronic material and device. For ease of comparison of our devices with other state-of-the-art devices reported in literatures, two tables containing the figures of merit for different infrared photodetectors and nonvolatile optoelectronic memories are incorporated. As shown in the Manuscript, our devices own simpler device configuration and better performance. For detail, please see Table S1 and S2 (page 31) in our revised Supplementary Information.

As listed below, the reviewer's concerns are well addressed, and have been incorporated in the revised Manuscript.

Comment 1: *The abstract indicates the electronic memory and IR sensor results were obtained for devices with individual nanosheets (this single layer of MoSSe?), but in the results section, the authors indicate that all data shown in the text is for a 7.6 nm sheet device.*

Our reply: We are sorry for causing this misunderstanding. Actually, the term “individual nanosheets” does not refer to “single layer of $\text{MoS}_{2x}\text{Se}_{2(1-x)}$ ”. The reason why the term “individual nanosheets” was adopted, is to distinguish our device structure from the current reported optoelectronic memories. As mentioned in the Manuscript, the structures of the current reported optoelectronic memories are usually very complicated, as they need the aid of additional charge transfer/storage medium (such as hetero-stacks, floating gate and functional substrates). By contrast, through introducing a robust trap centers, our multifunctional devices only need one channel material. Based on your comment, the corresponding descriptions have been revised from “individual nanosheet” to “individual TMD flake”.

Comment 2: *The anomalous temperature dependence of the FET I_{ds} - V_{gs} curves (decreasing I_{off} from 300 K to 180 K but increasing I_{off} from 140K to 80K, with an abrupt change in behavior between 140 and 180 K) needs to be better understood and justified with experimental data, such as any leakage gate/substrate currents that may*

be present (not shown). The proposed mechanism may be a plausible explanation for the anomalous behavior but is not supported by experimental data.

Our reply: The possibility that “the anomalous temperature dependence of the FET I_{DS} - V_{GS} curves comes from substrate (such as leakage gate/substrate current)” can be excluded via incorporating experimental data. Detailed discussions are given below.

(a) The leakage currents (I_{GS}) of device #1 above and below 140 K are provided in **Fig. R17a**. The leakage currents do not affect the source-drain currents (I_{DS}), and do not change significantly regardless of temperatures. Moreover, the current on/off ratio increases from 10 at dark state to $\sim 10^8$ under illumination with 1550 and 1940 nm lasers (Fig. 3a). It suggests that the anomalous behavior is not caused by leakage gate/substrate currents. The leakage currents of the other two devices at low temperatures, which are far less than their corresponding I_{DS} , are also displayed in **Fig. R17b, c**. (b) Furthermore, a number of devices with various thicknesses were fabricated on both SiO_2 and h-BN substrates (Fig. 2d, Supplementary Fig. 15 and 16), and exhibit similar electronic transport behavior. Thus, we believe that this anomalous temperature dependence is substrate-independent, repeatable and valid. (c) The temperature-dependent Raman spectra show that there is neither characteristic peak shift nor appearance of new peaks in the process of changing temperatures, eliminating the possible effects from structural phase transition (Supplementary Fig. 14). In view of the above, we ascribe this anomalous behavior to the intrinsic property of material. Based on your comments, more discussions about leakage current have been added to page 10 (lines 6-8) of the revised manuscript and Fig. S17 in Supplementary Information.

Figure R17 | The leakage current and source-drain current of our devices at low temperature.

Supported by density functional theory (DFT) calculations and electrical transport characterizations, we proposed the model of holes (minority carriers) trapping and detrapping for explaining the physical mechanism of the aforementioned phenomenon. At low temperature, due to the weak thermal perturbation, holes can be trapped in the defect levels more stably. These trapped holes with positive charge can effectively screen the effect of negative gate voltage (V_{GS}) and suppress the depletion of electrons in the channel, generating an anomalous high-current state at large negative V_{GS} . The corresponding schematic band diagrams are given in **Fig. R18**. The proposed model is further supported by (a) the photoresponse measurement with infrared lasers (1550, 1940 and 2699 nm) and (b) time-resolved current measurement under various temperatures (at $V_{GS} = -80$ V), which demonstrate the existence of in-gap traps and the influence of temperature on de-trapping process. Detailed discussions are given below.

Figure R18 | Schematic band diagrams of $\text{MoS}_{2x}\text{Se}_{2(1-x)}$ devices at $V_{GS} < 0$ V, $V_{DS} > 0$ V. Electrons and holes are represented by yellow and red balls, respectively. Black dashed lines in the bandgap denote the empty state (with trapped holes).

(a) Photo-excitation can effectively determine the state and position of the trap

levels in semiconductors (*Nano Lett.* 15, 5875, 2015). **Figure 3b** (see **Fig. R19a** below) shows the time-resolved photoresponse measurement at $V_{GS} = -80$ V. Under the illumination of our device with 1550 and 1940 nm lasers, the currents reduce directly to the order of $\sim 10^{-13}$ A. And the currents keep in that value (no photoresponse) even the laser is turned on again. The corresponding band diagrams are given in **Fig. R19b**. The negative photoresponse ($I_{\text{illumination}} < I_{\text{dark}}$) manifests the existence of in-gap traps that have captured holes (**Process I**). This is because the low-energy photons of 1550 and 1940 nm (0.8 eV and 0.64 eV) can only pump electrons from the valence band to the trap states in the bandgap (**Process II**). Accordingly, due to the recombination between the trapped holes and photo-excited electrons (i.e., the trapped holes are erased), the corresponding charge shielding effect vanishes. As a result, electrons in the conduction band are depleted by the large negative V_{GS} . Besides, the only once negative photoresponse indicates that the infrared lasers (1550 and 1940 nm) cannot trigger an interband light absorption again. This is because the holes could not be captured by the defect levels again at $V_{GS} = -80$ V due to the strong electrostatic attraction between the large negative V_{GS} and positive charges. Thus, in the following several photo-switching cycles (**Process III**), the defect levels are in the occupied state (without trapped holes), and electrons in the valence band could not be excited to these defect levels. And the infrared lasers (1550 and 1940 nm) cannot also trigger a band-edge light absorption owing to their insufficient energy (0.8 and 0.64 eV). Further, as shown in Supplementary Fig. 23 (see **Fig. R19c** below), under the illumination with 2699 nm and 3000 nm (0.46 eV and 0.41 eV) lasers, there is no negative or positive photoresponse for our device. It suggests the photon energy of 2699 nm (0.46 eV) is lower than the position of most trap levels. Therefore, we estimated that trap distribution should mainly concentrate in the range of 0.46 eV and 0.64 eV above the valence band edge.

Figure R19 | (a) Time-resolved photoresponse measurement at $V_{GS} = -80$ V. The blue rectangle represents the illuminated state. (b) The schematic band diagrams of $\text{MoS}_{2x}\text{Se}_{2(1-x)}$ at $V_{GS} = -80$ V under dark and illumination (1550 and 1940 nm) states. Electrons and holes are represented by yellow and red balls. Gray balls are the disappeared electrons or holes. Black dashed and solid lines in the bandgap denote the empty state (with trapped holes) and occupied state (without trapped holes). (c) I_{DS} - V_{GS} curves of the device #1 under dark and illuminated states with 2699 nm (0.46 eV) and 3000 nm (0.41 eV) lasers, $V_{DS} = 1$ V, $T = 80$ K.

(b) The effect of thermal perturbation on holes' de-trapping process is characterized by de-trapping time (τ_t): $\tau_t^{-1} = s_p N_v v_{th} \exp(-\Delta E/(kT))$, where s_p , N_v , v_{th} , ΔE , and k are the capture cross section of the trap center, the effective density of states in the valence bands, the thermal velocity of the carriers, the energy difference between the trap state and the valence band edge, and the Boltzmann constant, respectively. At a high temperature, the provided thermal energy (kT) can promote the de-trapping process by

thermally exciting the trapped holes back into valence band. This rapid process makes it difficult to observe the effects of trapping holes. With the decrease in temperature, it will be difficult to thermally excite the trapped holes back to the valence band due to the low thermal energy. It suggests that the effect of thermal perturbation on holes' de-trapping process starts to weaken. As a result, the de-trapping time increases exponentially with the decrease of temperature. At this point, we can easily detect the anomalous electronic transport properties induced by the trapped holes. The time-resolved current measurement under various temperatures ($V_{GS} = -80$ V) demonstrates the temperature-dependent holes' de-trapping process. As shown in **Fig. 2c** (see **Fig. R20** below), the anomalous high-current state at $V_{GS} = -80$ V (140 K) decreases gradually as the test time increases, corresponding to the gradual de-trapping process. Further, when the temperature drops to 80 K, due to the lower thermal energy, the de-trapping process is completely repressed and the high-current state at -80 V is very stable. These results can help us to determine the optimum operating temperature of the device, which is of vital importance to their practical application.

Figure R20 | Time-resolved current measurement under various temperatures. $V_{GS} = -80$ V, $V_{DS} = 1$ V.

According to your suggestion, more detailed explanations have been added into the “**Anomalous temperature dependence of electronic transport**” section of our revised Manuscript. For details please see the marked text on page 8-9 in our revised

Manuscript and Fig. S13, S18, S23 in the revised Supplementary Information.

Comment 3: *It is not clear why retention would be characterized at high field ($V_g = -80$ V) and 80 K when retention usually refers to the stability of an undisturbed state, under equilibrium, normal conditions.*

Our reply: The operating mode of a memory device is determined by its component materials and device configuration. For nonvolatile memory applications, a device should be bistable, that is, stable ON and OFF states should be achieved at the same gate condition. In order to increase the reliability of the memories, the “dynamic range” between the extreme OFF and ON states (at fixed V_g) should be maximized (*Nat. Nanotech.* 11, 769, 2016). Hence, for a specific optoelectronic memory, a specific operating condition and operating mode is very important and necessary. Examples include P3HT/DAE-Me optoelectronic memory operates at $V_g = -60$ V (*Nat. Nanotech.* 11, 769, 2016), WSe₂/BN optoelectronic memory operates at $V_g = 50$ V (*Nat. Commun.* 9, 2966, 2018), MoS₂/functionalized-SiO₂ optoelectronic memory operates at $V_g = 20$ V (*Nat. Commun.* 8, 14734, 2017), MoS₂/PbS optoelectronic memory operates at $T = 80$ K (*Sci. Adv.* 4, eaap7916, 2018).

As for our nonvolatile infrared memory, its bistable behavior is based on the stable trapping of holes and their release in an individual TMD flake, which is totally different from the conventional nonvolatile memories. Its operating conditions are determined by its transport and photoresponse properties. As shown in Fig. 3a (see **Fig. R21a** below), the maximum difference (10^8 , at least two orders of magnitude higher than the reported optoelectronic memories) between the ON (dark) and OFF (infrared illuminated) states was observed under $V_{GS} = -80$ V and 80 K. And with this condition, the holes’ de-trapping process is completely suppressed, and the high-current state maintained very well (see **Fig. R20** above). In addition, as shown in the new Fig. 4c (see **Fig. R21b** below), both the ON and OFF states can be maintained very well, indicating that using this measurement condition won’t disturb the reading state after being programmed and erased. Hence, to increase the reliability

of our memories, using this measurement condition is reasonable. The study of transport and photoresponse properties can also help us to determine the optimum operating condition of the device, which is of vital importance to their practical application.

Based on your comments, we have added the corresponding text to the revised Manuscript as “As shown in Fig. 3a, a maximum difference (10^8) between the ON (dark) and OFF (infrared illuminated) states was observed under $V_{GS} = -80$ V and $T = 80$ K, implying two distinctly different memory states.” Please see the marked text on page 16, lines 23-25 in our revised Manuscript.

Figure R21 | (a) I_{DS} - V_{GS} curves of the device under dark and illuminated states with 473 nm (248 mW cm^{-2}), 1550 nm ($100.6 \text{ } \mu\text{W cm}^{-2}$), and 1940 nm ($327.1 \text{ } \mu\text{W cm}^{-2}$) lasers. $V_{DS} = 1$ V, $T = 80$ K. (b) Retention performance of two memory states after electrically programming and optically erasing.

Comment 4: *The proposed write/erase/read schemes are not clear, and why one electrical and one optical. The reason for these (electrical write and optical erase?) should be justified. The reason for reading at high gate field of -80 V is also not clear since reading is also typically done under low, non-disturbing gate and s/d fields.*

Our reply: As discussed in **Reply 3**, the bistable behavior of our nonvolatile infrared memory is based on the stable trapping of holes and their release in an individual TMD flake, which is totally different from the conventional nonvolatile memories.

For a better illustration, we first discuss the question “*why one electrical and one optical. The reason for these (electrical write and optical erase?) should be justified*”, then reply to the comment “*The proposed write/erase/read schemes are not clear*”. As for the comment “*The reason for reading at high gate field of -80 V is also not clear since reading is also typically done under low, non-disturbing gate and s/d fields*”, it has been detailedly discussed in our **Reply 3**.

1. why one electrical and one optical. The reason for these (electrical write and optical erase?) should be justified.

The operating mode of a memory device is determined by its component materials and device configuration. As silicon-based flash memories are approaching their fundamental limit, novel materials and innovative device concepts are being investigated. Recently, benefiting from its high capacity, low energy consumption and long-distance secure communications, optical data storage has been considered as a highly promising alternative to conventional approaches (*Nat. Rev. Mater.* 1, 16070, 2016; *Science* 357, 1392, 2017). Specifically, the infrared radiations of 850, 1310 and 1550 nm with low energy dissipation in optical fibers are typical optical communication wavebands. Therefore, the optoelectronic devices that would convert and store infrared information into electrical signals, thereby enabling optical data communications, are highly pursued.

As for our nonvolatile memories, as illustrated in **Fig. R22**, no obvious hysteresis (i.e., memory window) was observed along the two different sweeping directions of gate voltage (V_{GS}), suggesting that only using the electrical pulse cannot define “identifiable bistable states”. Thus, a specific operating mode (electrical programming and optical erasing) is necessary for our infrared memories, just like all other reported 2D optoelectronic memory devices (*Nat. Commun.* 9, 2966, 2018; *Nat. Commun.* 8, 14734, 2017; *Sci. Adv.* 4, eaap7916, 2018; *Adv. Mater.* 31, 1807075, 2019; all of them must be electrically erased or programmed; 2D optoelectronic memory with all-optical switching has not yet been realized). And compared to these reported optoelectronic memories, our devices own better performance and simpler device

configuration (just based on an individual TMD flake, no need the aid of additional charge transfer/storage medium). These have been detailedly discussed in the Manuscript. For better comparison, a table containing the figure of merits for different nonvolatile optoelectronic memories has been added into our revised Supplementary Information as Table S2 (page 31).

Figure R22 | I_{DS} - V_{GS} curve of the device along two different V_{GS} sweeping directions under dark state. No obvious hysteresis (i.e., memory window) was observed.

2. The proposed write/erase/read schemes are not clear.

To make it easy to read, we have re-plotted the schematic of the program, erase and read process of a single operation cycle. The corresponding descriptions have also been revised. Please see the new **Fig. 4b** (**Fig. R23a** below), and the marked text on page 17 (lines 2-14) in our revised Manuscript. They are also provided below.

(Process I, Program) The programming operation is achieved by applying a nonnegative gate voltage ($V_{GS} \geq 0$ V) to the devices. In this scenario, the corresponding electrostatic attraction between the gate voltage and positive charges disappears, and the capture of holes becomes much easier. At the same time, non-equilibrium electron concentration (Δn) increases due to electrostatic electron-doping. **(Process II, Readout under the program state)** These trapped holes can be retained even after removing the corresponding positive V_{GS} , thereby acting as positive charge centers localized in the bandgap. Benefiting from their effective shielding effect to negative gate field (suppress the depletion of electrons in

the channel), a retentive low resistance state (LRS) can be read out at $V_{GS} = -80$ V (**Region II in Fig. R23b**).

(**Process III, Erase**) To erase the stored holes, infrared laser pulse (1550 and 1940 nm) is applied under $V_{GS} = -80$ V. In this scenario, the photo-excited electrons can recombine with the stored holes to eliminate the charge shielding effect. (**Process IV, Readout under the erase state**) As a consequence, in the absence of charge shielding centers, the electrons in the channel will be depleted by the large negative gate voltage, and the device goes back to a retentive high resistance state (HRS, read out at $V_{GS} = -80$ V).

Figure R23 | (a) Schematic band diagrams of the corresponding program (I), readout (II, IV), and erase (III) operation. Electrons and holes are represented by yellow and red balls. Gray balls are the disappeared electrons or holes. Black dashed and solid

lines in the bandgap denote the empty state (with trapped holes) and occupied state (without trapped holes). (b) Time evolution of current in a single operation cycle with various programming voltages (V_{pro}). $V_{\text{DS}} = 1$ V, $T = 80$ K. Region I, II, III, and IV denote the states of programming, readout (under program state), erasing, and readout (under erase state) shown in (a), respectively.

Comment 5: *The negative (only once) and typical positive photoresponse are confusing, and not consistent with the model proposed for the FET anomalous temperature dependence behavior, based on the minority carrier trapping and detrapping.*

Our reply: The reason why there is a negative photoresponse (only once) under illumination with 1550 and 1940 nm lasers has been detailedly discussed in the previous reply. Please see **Fig. R19b** and the third paragraph in our **Reply 2**. As for the typical positive photoresponse ($I_{\text{illumination}} > I_{\text{dark}}$) of our device, it occurs under the illumination with 473 nm laser, for its photon energy (2.62 eV) is larger than the bandgap of $\text{MoS}_{2x}\text{Se}_{2(1-x)}$. Detailed discussions are given below.

Fig. R24a (blue line) shows the time-resolved photoresponse measurement (at $V_{\text{GS}} = -80$ V) under the illumination with 473 nm laser. As discussed in our **Reply 4**, the trapped holes with positive charge can effectively screen the negative gate voltage and suppress the depletion of electrons, generating a high-current state at $V_{\text{GS}} = -80$ V (Process I in **Fig. R24b**). Under the first illumination with 473 nm laser, the current dropped and remained at a constant value ($\sim 10^{-9}$ A). And when the laser was switched off, the current reduced again to the order of $\sim 10^{-13}$ A. In contrast, under the illumination with 1550 and 1940 nm lasers (pink and red lines), the current reduced directly to $\sim 10^{-13}$ A even the laser was turned on again.

This difference comes from the different photon energies of lasers. The low-energy photons of 1550 and 1940 nm (0.8 and 0.64 eV) can only pump electrons from the valence band to the in-gap trap states, and cannot pump them to the conduction band. Thus, when turning on the laser, the current reduced directly to the order of $\sim 10^{-13}$ A

(Details please see the third paragraph in our **Reply 2**). However, the high-energy photons of 473 nm laser (2.62 eV, larger than the bandgap of $\text{MoS}_{2x}\text{Se}_{2(1-x)}$) can excite electrons not only to the in-gap trap levels but also to the conduction band (Process II in **Fig. R24b**). It means that the photoresponse with 473 nm laser is a result of the combined effect of interband and band-edge light absorption. Therefore, even if the trapped holes have been erased (via recombining with the photo-excited electrons), the device could not return to a high resistance state (HRS) under the illumination with 473 nm laser. But when the 473 nm laser was switched off, the current can reduce to $\sim 10^{-13}$ A due to the vanishing charge shielding effect and band-edge light absorption.

Because the holes could not be captured by the defect levels again at $V_{\text{GS}} = -80$ V due to the strong electrostatic attraction between the large negative V_{GS} and positive charges. Thus, in the following several photo-switching cycles (Process III in **Fig. R24b**), the defect levels are in the occupied state (without trapped holes), and electrons in the valence band could not be excited to these defect levels. Subsequently, the device exhibits similar photoresponse properties to that of MoS_2 : that is, no response to infrared lasers (1550 and 1940 nm), but exhibit a normal photoswitching behavior (i.e. positive photoresponse) with the switch of 473 nm laser.

These unique photoresponse properties are exactly consistent with our model and experimental data. According to your comments, the corresponding descriptions have been modified based on the above discussions. Please see the marked text on page 12-13 in our revised Manuscript and Fig. S18 in Supplementary Information.

Figure R24 | (a) Time-resolved photoresponse measurement at $V_{GS} = -80$ V. The blue rectangle represents the illuminated state. (b) The schematic band diagrams of $\text{MoS}_{2x}\text{Se}_{2(1-x)}$ at $V_{GS} = -80$ V under dark and illumination (473 nm) states. Electrons and holes are represented by yellow and red balls. Gray balls are the disappeared electrons or holes. Black dashed and solid lines in the bandgap denote the empty state (with trapped holes) and occupied state (without trapped holes).

Comment 6: *The use of deep states for minority carrier trapping for both memory and sensing operation is confusing and also not consistent with shown I_d - V_{gs} n-channel FET operation (which would require a p-type semiconductor channel to be inverted, in which holes would be the majority carriers, not the minority). Or holes would come from substrate under high positive gate field? Very minimal tunneling is expected through 280 nm oxide under 80 V (0.28 V/nm).*

Our reply: Both non-volatile memory and infrared detection are based on the robust trap effect, which can capture holes (minority carriers) and store them steadily. In **Reply 2**, based on the experimental data (leakage current measurement, reliable reproducibility on various substrates, temperature-dependent Raman spectra), we have excluded the possibility that “the anomalous temperature dependence of the FET I_{DS} - V_{GS} curves comes from the leakage current”, and confirmed that this anomalous property is the intrinsic property of the material. Furthermore, supported by density functional theory calculations and electrical transport characterizations, we proposed

the model of holes (minority carriers) trapping and detrapping for explaining the physical mechanism of such a phenomenon. And this physical model is exactly consistent with the experimental results of photoresponse measurement and time-resolved current measurement under various temperatures.

Actually, for bare low-dimensional semiconductors (no hybrid structures), the filling of trap states is involved with the Fermi level, thus n-type channel mainly has hole-trap states while p-type channel mainly has electron-trap states (*Adv. Sci.* 4, 1700323, 2017). In our case, although the device exhibits an undoubted n-type feature, the minority carriers (holes) must exist in the channel material to maintain the charge neutrality. More importantly, under non-equilibrium conditions (such as light or electricity excitation), trap effect refers to the effect of defect levels to accumulate non-equilibrium (excess) carriers. These excess minority carriers have more significant influence on the carrier transport than excess majority carriers, because the amount of excess majority carriers is far less than the equilibrium majority carriers (JP Colinge, CA Colinge. *Physics of semiconductor devices*[M]. Springer Science & Business Media, 2005). Hence, an extra source of holes (such as p-type semiconductors) is not needed.

Taking n-type semiconductor as an example (schematic diagram in **Fig. R25**), it has an upper Fermi level. The deep levels can't capture electrons but only capture holes (minority carriers). This is because the traps that can capture electrons are usually exposed above the Fermi level (corresponding to the shallow level). Even if the electrons are captured, they are easily thermally excited back to the conduction band. In addition, after capturing holes, the deep in-gap levels (empty states: with trapped holes) provide a nature platform to accept electrons from the valence band excited by infrared (Process II in **Fig. R25**). Therefore, the proposed minority carrier trapping for non-volatile memory and infrared detection is theoretically rational.

Figure R25 | Schematic band diagrams of deep traps in a typical n-type semiconductor. Process I, excess minority carriers (holes) will be captured and stored in *deep traps* under non-equilibrium conditions. Process II, the trapped holes can be released through recombining with the light-excited electrons from the valence band. Electrons and holes are represented by blue and red circles with e^- and h^+ , respectively.

Based on your critical suggestion, although the consistency between density functional theory calculations, semiconductor theory and experimental data has been demonstrated, we will elaborate several experimental facts to further eliminate the concerns of reviewer. The corresponding descriptions in the Manuscript have also been revised accordingly. Please see the marked text on page 8-10 in our revised Manuscript.

(a) As shown in **Fig. R26**, the depletion of electrons is completely suppressed at $V_{GS} = -80$ V. It suggests that the large negative gate voltage is screened and attenuated by the trapped positive charges, which is consistent with the model of holes' trapping. Actually, the screen effect of trapped carries to the opposite gate electric field has been widely reported (*Nat. Commun.* 9, 2966, 2018; *Nat. Nanotech.* 8, 826, 2013; *Nat. Commun.* 4, 1624, 2013; *Nat. Commun.* 8, 14734, 2017). That is, the trapped positive charges can screen the effect of negative gate voltage while the trapped negative charges can screen the effect of positive gate voltage.

Figure R26 | I_{DS} - V_{GS} curves of device #1 under 300 K and 80 K. $V_{DS} = 1$ V.

(b) For device with a good ohmic contact under 300 K and 80 K (**Fig. R27a, b**), the two-terminal conductivity is defined as $\sigma = I_{DS}/V_{DS} \times L/W$ (*ACS Nano* 11, 7362, 2017) and displayed as a function of temperature under various V_{GS} (**Fig. R27c**). The charge transport is thermally activated at $160 \text{ K} < T < 300 \text{ K}$ based on the good agreement of the data with the activation transport model: $G(T) = G_0 \exp(-E_a/k_B T)$, where k_B is the Boltzmann constant, E_a is the activation energy, and G_0 is the fitting parameter (*Nat. Mater.* 12, 815, 2013; *Nano Lett.* 16, 6383, 2016). The activation energy, specifying the energy difference between the Fermi level and the mobility edge, is extracted and analyzed with the thermally activated transport. As shown in **Fig. R27d**, the activation energy is dependent on gate voltage, and is lower than that of other 2D materials on SiO_2 substrate (*Nano Lett.* 16, 6383, 2016; *ACS Nano* 8, 4859, 2014). The value of E_a is ~ 1 meV at $V_{GS} = -40$ V. Such low E_a value verifies the proposed model of holes' trapping and de-trapping for explaining the anomalous electronic behavior. Because the Fermi level (E_f) is very close to the mobility edge or the conduction band minimum (*ACS Nano*, 5, 7707, 2011), the unoccupied defect levels upon E_f are very shallow, and difficult to trap electrons stably even under low temperature (Process I, schematic diagram in **Fig. R25**). By contrast, the deep defect levels below E_f can trap and store holes more stably at low temperature, due to the weak thermal perturbation. These trapped holes with positive charge can effectively screen the effect of negative gate voltage (V_{GS}) and suppress the depletion of electrons in the channel, generating

the anomalous high-current state at negative V_{GS} .

Figure R27 | (a, b) I_{DS} - V_{DS} curves under various V_{GS} from -30 V to -70 V. $T = 300$ K and 80 K. (c) The temperature and gate voltage dependence of conductivity curves of the device. (d) E_a curve as a function of V_{GS} .

(c) We have demonstrated that the anomalous temperature dependence of the FET I_{DS} - V_{GS} curves is not coming from the substrate leakage current (**Reply 2**). And in this section, based on the gate voltage-dependent memory effect (**Fig. R28**), we can completely exclude the possibility that “holes would come from substrate under high positive gate field”. As discussed below, we show that the capture of holes can occur when a small gate voltage (even 0 V) is applied for programming. In this scenario, it is difficult for holes to tunnel through 280 nm SiO₂.

Figure R28 | Time evolution of current in a single operation cycle with various programming voltages (V_{pro}). $V_{DS} = 1$ V, $T = 80$ K. Region I, II, III, and IV denote the states of programming, readout (under program state), erasing, and readout (under erase state), respectively.

To better explain the gate voltage (V_{GS})-dependent memory effect, the schematic diagrams of various programming and read processes are displayed in Fig. S27 of the revised Supplementary Information (see **Fig. R29** below). As discussed in the manuscript, holes can be stored into trap levels and retained after applying an appropriate V_{GS} pulse to the device. These trapped holes can act as charge shielding centers and provide sufficient shielding effect to the gate field of -80 V, thereby suppressing the depletion of electrons in the channel. Thus, a retentive low resistance state (LRS) can be read out at $V_{GS} = -80$ V. As can be seen from region II in **Fig. R28**, by applying different programming V_{GS} (V_{pro}), we can get different readout states.

When V_{pro} is set in the range of -80 to -30 V (**Fig. R29a**), the strong electrostatic attraction between the large negative gate voltage and positive charges is not conducive to capture and store holes (i). Thus, in the absence of charge shielding centers, the electrons in the channel are depleted by large negative gate voltage, and the LRS (measured at $V_{GS} = -80$ V, $V_{DS} = 1$ V) cannot be effectively identified (ii).

Along with the increase of V_{pro} (-30 V $< V_{pro} < 0$ V, **Fig. R29b**), the LRS can be easily read out and gradually becomes stable. It suggests that holes start to be

captured by defect levels (iii), which can provide stronger shielding effect to the gate field of -80 V (iv). Hence, $V_{\text{pro}} = -30$ V is a threshold voltage (V_T) for storing charges here.

Further, when a nonnegative V_{pro} ($V_{\text{pro}} > 0$ V or $V_{\text{pro}} = 0$ V) is applied (Fig. R29c), the corresponding electrostatic attraction disappears, and the capture of holes becomes much easier (v, vi). Sufficient numbers of trapped holes give rise to a robust charge shielding effect (vii). Thus, very stable LRS can be obtained at $V_{\text{GS}} = -80$ V.

Figure R29 | The schematic diagrams of the programming processes (i, iii, v and vi) under various programming V_{GS} (V_{pro}) and the corresponding LRS-read processes (ii, iv and vii) at $V_{\text{GS}} = -80$ V. (a) $-80 \text{ V} < V_{\text{pro}} < V_T$; (b) $V_T < V_{\text{pro}} < 0 \text{ V}$; (c) $V_{\text{pro}} \geq 0 \text{ V}$. Electrons and holes are represented by yellow and red balls, respectively. Gray balls

are the disappeared electrons and holes. Black dashed and solid lines in the bandgap denote the empty state (with trapped holes) and occupied state (without trapped holes), respectively. Black arrows represent the trapping holes.

Comment 7: *Very little detail is provided on the fabrication and structure of the devices. What are the dimensions, L , W of the FETs, what type and level doping is the Si substrate which acts as the back-gate of the FETs, what type of oxide is on the substrate, thermally grown or deposited? What are the intrinsic electronic properties of the semiconductor material (TMD flake), conductivity, mobility, carrier type and carrier concentration? These are important to interpret the I_d - V_{gs} curves provided.*

Our reply: Thanks for bringing this issue to our attention. More details about the fabrication and structure of our devices have been provided. The Si substrate acting as the back gate of the FETs is n-doped with phosphorus. Its resistivity is 1-30 $\Omega \cdot \text{cm}$. And the oxide on the substrate is thermally grown. The substrates were cleaned before use, with a mixed solution of H_2O_2 and H_2SO_4 (volume ratio is 3:7) for 2 h. The source/drain electrodes (10/50 nm Cr/Au) were fabricated by standard electron-beam lithography (Nova200 NanoLab) and thermal evaporation apparatus (with a rate of 0.05 \AA s^{-1}). Finally, a standard lift-off process was adopted. These information have been added to the “**Methods - Device fabrication and characterization**” section of the revised Manuscript (page 21). The dimensions of all the FETs displayed in the Manuscript have also been provided. For details, please see the caption of Fig. S15 in revised Supplementary Information.

In our research, all of the devices show prominent n-type feature, and their field effect mobilities have been summarized in Fig. 2e. Besides, two-dimensional carrier concentration (n_{2D}) at the MOSFET can be calculated by the equation: $n_{2D} = C_{\text{ox}}|V_{\text{GS}} - V_{\text{TH}}|/e$ (*Nat. Mater.* 12, 815, 2013; *Nat. Commun.* 4, 2642, 2013; *Nat. Commun.* 6, 6485, 2015), where C_{ox} is gate capacitance and V_{TH} is the threshold voltage extracted by linear extrapolation of I_{DS} - V_{GS} characteristics in the linear region. For our $\text{MoS}_{2x}\text{Se}_{2(1-x)}$ -based FETs, the majority carrier concentration (electrons) is in

the range of $6\sim 8 \times 10^{-12} \text{ cm}^{-2}$ at $V_{\text{GS}} = 80 \text{ V}$. The field effect mobility and carrier concentration of our FETs are similar to that of reported MoS₂-based back-gated FETs (*Chem. Soc. Rev.* 45, 118, 2016; *Nat. Commun.* 4, 2642, 2013; *Nat. Mater.* 12, 815, 2013), which are not main factors for the anomalous temperature dependence of $I_{\text{DS}}-V_{\text{GS}}$ curves. The role of carrier type, conductivity and activation energy in the anomalous temperature dependence of $I_{\text{DS}}-V_{\text{GS}}$ curves have also been elaborated. Please see our **Reply 6**.

Comment 8: *The article is, in general, well written and organized but requires some English editing throughout. For example, the abstract starts with ‘Defect plays’ instead of ‘defects play’ which would be the common usage in the field.*

Our reply: We have now double checked the manuscript and made corrections for the mistakes. In addition, the paper has now been edited by a native English speaker who has also a strong scientific background. Now we believe it has got much improved.

Comment 9: *The data plots are generally too small and difficult to read.*

Our reply: Many thanks for this kindly reminder. We have re-plotted the figures to make it easy to read. Please see the new figures in the revised Manuscript and Supplementary Information.

Reviewers' comments:

Reviewer #1 (Remarks to the Author):

In response to my questions, the author explained them one by one through supplementary experiments or references. I think this version of the article meets the journal's acceptance requirements, so I recommend this article to be accepted.

Reviewer #2 (Remarks to the Author):

The authors have substantially improved the manuscript by incorporating the suggestions and concerns raised by me. They have performed some additional experiments to further support their claims. The work is novel and can be published now in Nature Communications. No further review is needed from my part.

Reviewer #3 (Remarks to the Author):

I think there is still no evidence for the conclusions of this work - anomalous temperature dependence and photo-response - and additional characterization is required.

The authors show Igs to exclude the possibility of leakage currents explaining some of the anomalous behaviors. All the currents need to be shown, especially I_d and I_s , to check if they have the same magnitude. There may be leakage current between the gate and drain contacts.

The band diagrams explaining the write, erase and read mechanisms are confusing and seem to have problems. The negative gate bias would bend the semiconductor bands upward, not downward at the semiconductor-insulator interface, along the G-channel direction (as shown for the S-D direction). It is not clear what is being shown.

The hysteresis curve shown, Fig. R22, to exclude the effect of normal charge trapping, is not conclusive as the device is always 'on' and the current is very large; in order to check charge trapping hysteresis the I_d - V_g should be plotted in log scale and the device should have a clear threshold voltage (not clear in this case). Were these characteristics measured at room temperature? What is the drain voltage?

For practical memory properties, retention characteristics must be determined for an 'undisturbed' device, with no fields applied, and at room temperature, not 80 K. I think retention at low temperature and under high field is not meaningful.

Reviewer #1

In response to my questions, the author explained them one by one through supplementary experiments or references. I think this version of the article meets the journal's acceptance requirements, so I recommend this article to be accepted.

Our reply: We would like to thank you for reviewing our paper, we appreciate your insightful comments on our research.

Reviewer #2

The authors have substantially improved the manuscript by incorporating the suggestions and concerns raised by me. They have performed some additional experiments to further support their claims. The work is novel and can be published now in Nature Communications. No further review is needed from my part.

Our reply: We would like to thank you for reviewing our paper, we appreciate your insightful comments on our research.

Reviewer #3

I think there is still no evidence for the conclusions of this work - anomalous temperature dependence and photo-response - and additional characterization is required.

Our reply: We thank you for spending time to carefully review our work and giving your thoughtful comments. In this Letter, based on your suggestion, we will explicitly give some crucial experimental evidences again to support our conclusions. It should be noted that “the anomalous temperature dependence of electronic transport, and the related photo-response and infrared memory effect” are **objective experimental facts**,

not the extracted conclusions.

Firstly, we excluded the possibility that the anomalous property comes from the substrate, and confirmed that this anomalous property is the intrinsic defect property of the material. This is based on these four experimental evidences: (a) **a number of devices with various thicknesses and substrates** exhibit very similar electronic transport behavior, indicating this anomalous temperature dependence is substrate-independent, valid and highly repeatable (Fig. 2d in the Manuscript, Fig. S15 and S16 in Supplementary Information, also described in Reply 2 to Reviewer #3 in our previous Response Letter); (b) **detailed leakage current measurements** suggest that the anomalous behavior is not caused by the leakage current between the gate and drain/source contacts (Fig. S17 and S18 in Supplementary Information, also described in Reply 1 to Reviewer #3 in this Letter); (c) **gate voltage-dependent memory effect measurements** suggest that the holes are not coming from the high gate field (Fig. 4e and 4f in the Manuscript, Fig. S28 and S29 in Supplementary Information, also described in Reply 6 to Reviewer #3 in our previous Response Letter); (d) **temperature-dependent Raman spectra** eliminate the possible effects from structural phase transition (Fig. S14 in Supplementary Information, also described in Reply 2 to Reviewer #3 in our previous Response Letter).

Furthermore, supported by **density functional theory prediction** for the trap formation (both hybrid functional and PBE functional are adopted) and electrical transport characterizations, we proposed the model of holes (minority carriers) trapping and de-trapping to explain the underlying physical mechanism. The rationality of this model is proved by four experimental evidences: (a) **detailed wavelength-dependent photo-response study** (473, 808, 1550, 1940, 2699 and 3000 nm) demonstrates the existence of in-gap traps and their distribution inside the bandgap (Fig. 3 in the Manuscript, Fig. S20, S22-27 in Supplementary Information, also described in Reply 2 to Reviewer #3 in our previous Response Letter); (b) **the suppressed depletion of electrons** at $V_{GS} = -80$ V demonstrates the large negative gate voltage is screened and attenuated by the trapped positive charges, which is

consistent with the model of holes' trapping (Fig. 2b in the Manuscript, Fig. S15 and S16 in Supplementary Information, also described in Reply 6 to Reviewer #3 in our previous Response Letter); (c) **time-resolved current measurement under various temperatures** demonstrates the temperature-dependent holes' de-trapping process (Fig. 2c in the Manuscript, also described in Reply 2 to Reviewer #3 in our previous Response Letter); (d) **detailed study of temperature-dependent conductivity and activation energy** indicate the upper Fermi level in the bandgap, causing that the deep defect levels are occupied and can trap holes (Fig. S12 in Supplementary Information, also described in Reply 6 to Reviewer #3 in our previous Response Letter).

Actually, these objective experimental facts, along with the density functional theory calculations, are the basis of our proposed physical model. They are coherent, mutually supporting, reliable and highly repeatable. We think they are sufficient to support our conclusions in the Manuscript.

As listed below, all your new concerns are well addressed, and have been incorporated in the revised Manuscript.

Comment 1: *The authors show I_{gs} to exclude the possibility of leakage currents explaining some of the anomalous behaviors. All the currents need to be shown, especially I_d and I_s , to check if they have the same magnitude. There may be leakage current between the gate and drain contacts.*

Our reply: Based on your comments, all the currents of a typical device were displayed in **Fig. S18** of the revised Supplementary Information (see **Fig. R1** below). Obviously, the drain current and source current have the same magnitude under various temperatures. Also, the gate current is always much lower than the drain current and source current. These results suggest that there is no leakage current between the gate and drain/source contacts, further confirming that the anomalous behaviors are the intrinsic defect property of material.

Figure R1 | All the currents of a typical device at various temperatures, plotted in both log (Top) and linear scales (Bottom).

Comment 2: *The band diagrams explaining the write, erase and read mechanisms are confusing and seem to have problems. The negative gate bias would bend the semiconductor bands upward, not downward at the semiconductor-insulator interface, along the G-channel direction (as shown for the S-D direction). It is not clear what is being shown.*

Our reply: It is true that the positive/negative gate bias would bend the

semiconductor bands downward/upward at the semiconductor-insulator interface. Actually, this is also the basis of our band diagrams. Detailed descriptions are given below.

As illustrated in **Fig. R2: (Process I, Program)** the programming operation is achieved by applying a nonnegative gate voltage ($V_{GS} \geq 0$ V) to the devices. In this scenario, the corresponding electrostatic attraction between the gate voltage and positive charges disappears, and the capture of holes becomes much easier. At the same time, non-equilibrium electron concentration (Δn) increases due to electrostatic electron-doping. Thus, we depicted the semiconductor band bending downward at the semiconductor-insulator interface. **(Process II, Readout under the program state)** These trapped holes can be retained even after removing the corresponding positive V_{GS} , thereby *acting as positive charge centers* localized in the bandgap. Benefiting from their effective shielding effect to negative gate field (suppress the depletion of electrons in the channel), the semiconductor band at semiconductor-insulator interface cannot change dramatically and a retentive low resistance state (LRS) can be read out at $V_{GS} = -80$ V. Thus, for readers to better understand the effect of shielding, we still depicted the semiconductor band slightly bending downward at the semiconductor-insulator interface. **(Process III, Erase)** To erase the stored holes, infrared laser pulse (1550 and 1940 nm) is applied under $V_{GS} = -80$ V. In this scenario, the photo-excited electrons can recombine with the stored holes to eliminate the charge shielding effect. **(Process IV, Readout under the erase state)** As a consequence, in the absence of charge shielding centers, the electrons in the channel will be depleted by the large negative gate voltage, and the device goes back to a retentive high resistance state (HRS, read out at $V_{GS} = -80$ V). Due to the vanishing charge shielding effect in the two processes, the semiconductor band bends upward at the semiconductor-insulator interface.

Based on your comment, we have added the corresponding description into the revised Manuscript as “Benefiting from their shielding effect to negative gate field (suppress the depletion of electrons in the channel), the semiconductor band at

semiconductor-insulator interface cannot change dramatically and a retentive low resistance state (LRS) can be read out at $V_{GS} = -80$ V.” Please see the marked text on page 17 (line 10-11) in our revised Manuscript.

Figure R2 | Schematic band diagrams of the corresponding program (I), readout (II, IV), and erase (III) operation. Electrons and holes are represented by yellow and red balls. Gray balls are the disappeared electrons or holes. Black dashed and solid lines in the bandgap denote the empty state (with trapped holes) and occupied state (without trapped holes).

Comment 3: *The hysteresis curve shown, Fig. R22, to exclude the effect of normal charge trapping, is not conclusive as the device is always 'on' and the current is very large; in order to check charge trapping hysteresis the I_d - V_g should be plotted in log scale and the device should have a clear threshold voltage (not clear in this case). Were these characteristics measured at room temperature? What is the drain voltage?*

Our reply: Based on your comments, the I_{DS} - V_{GS} curves (Fig. R22 in the previous Response Letter) are plotted in both linear and log scales. This figure is used to response to your previous Comment 4. As shown in **Fig. R3** below, no obvious hysteresis (i.e., memory window) was observed along the two different sweeping

directions of gate voltage (V_{GS}) (measured at $V_{DS} = 1$ V and $T = 80$ K), suggesting that only using the electrical pulse cannot define “identifiable bistable states”.

Figure R3 | I_{DS} - V_{GS} curve of the device along two different V_{GS} sweeping directions under dark state, plotted in linear (a) and log (b) scales. No obvious hysteresis (i.e., memory window) was observed.

Comment 4: *For practical memory properties, retention characteristics must be determined for an 'undisturbed' device, with no fields applied, and at room temperature, not 80 K. I think retention at low temperature and under high field is not meaningful.*

Our reply: The operating mode of a memory device is determined by its component materials and device configuration. For nonvolatile memory applications, a device should be bistable, that is, stable ON and OFF states should be achieved at the same gate condition. And “operated with no fields applied or at room temperature” is not the necessary condition for optoelectronic devices (especially for nonvolatile infrared memory). Actually, lots of practical devices are operated at low temperature and with fields. One of the most typical examples is widely used commercial HgCdTe infrared devices which operate at low temperatures. For a specific optoelectronic memory, a specific operating condition and operating mode is very important and necessary to identify the distinctly different and stable memory states (*Nat. Nanotech.* 11, 769,

2016; *Nat. Nanotech.* 8, 826, 2013; *Nat. Commun.* 9, 2966, 2018; *Nat. Commun.* 8, 14734, 2017; *Nat. Commun.* 5, 4720, 2014; *Sci. Adv.* 4, eaap7916, 2018; *Adv. Mater.* 30, 1706647, 2018; *Small* 15, 1804661, 2019, these have been described in Reply 3 to Reviewer #3 in our previous Response Letter).

In addition, our devices exhibit excellent retention performance (both the ON and OFF states exhibit no noticeable change over 10^4 s), repeatable switching behaviors and ultrahigh memory program/erase ratio ($\sim 10^8$), indicating that using this measurement condition won't disturb their retention characteristics. In the following, we will further describe the importance of our work.

As silicon-based flash memories are approaching their fundamental limit, novel materials and innovative device concepts are being investigated. Recently, benefiting from its high capacity, low energy consumption and long-distance secure communications, *optical data storage* has been considered as a highly promising alternative to conventional approaches (*Nat. Rev. Mater.* 1, 16070, 2016; *Science* 357, 1392, 2017). Specifically, *the infrared radiations* of 850, 1310 and 1550 nm with low energy dissipation in optical fibers are typical optical communication wavebands. Therefore, the optoelectronic devices that would convert and store infrared information into electrical signals, thereby enabling optical data communications, are highly pursued. In this regard, we designed a novel nonvolatile infrared memory device. Its bistable behavior is based on the stable trapping of holes and their release in an individual TMD flake.

For technical applications, this kind of simple configuration (just based on an individual TMD flake, no need the aid of additional charge transfer/storage medium) provides a new direction in simplifying device fabrication procedures and improving integration level. The device can simultaneously operate as an infrared detector and nonvolatile infrared memory. Besides, the investigation of high-performance devices in the extreme environment is particularly important for assessing the applicability of our materials for intended purpose. **For scientific research**, we observed the anomalous temperature dependence of electronic transport for the first time,

developed the applications of functional defects, and proposed an entirely new infrared detection and storage mechanism. Compared to those previously reported optoelectronic memories (Table S2 in Supplementary Information), our devices own simpler device configuration and much better performance (the program/erase current ratio and optical erasing time reach 10^8 and 0.9 ms, both are at least two orders of magnitude higher than the reported optoelectronic memories).

Based on your comments, more discussions about the value of our work have been added. For details please see the highlighted text in the “**Infrared memory and its operational mechanism**” section of our revised Manuscript.

Reviewers' comments:

Reviewer #3 (Remarks to the Author):

The manuscript includes detailed experimental characterization and simulation results, which can be useful to better understand these devices and materials but the proposed model for the anomalous transport and its dependence on temperature and illumination is still unclear and confusing. Why would minority carriers (holes) be captured into deep traps at the semiconductor interface under a large positive gate voltage that accumulates the semiconductor interface with electrons?

[In fig. 1: "Process I, excess minority carriers (holes) will be captured and stored in Et under non-equilibrium conditions."

In fig. 4 Process I, program, ($V_{gs} > 0$).]

The output characteristics shown in the supplement (S12 and S20) also do not look like those of FETs suggesting a different type of operation involved in these devices, perhaps resulting from specific fabrication conditions. Simpler MOS characterization of this gate stack (Cr/Au//SiO₂//MoSSe) is needed to interpret the FET device I-V characteristics shown here.

Reviewer #4 (Remarks to the Author):

I have carefully read the manuscript by Yin et al. and the latest review round. I find the responses by the authors and the additional experiments sufficiently convincing and can recommend the manuscript for acceptance but need some clarifications

In Figure 1d the endurance is provided as 120 cycles. This is quite poor endurance. Could the authors comment on this?

In how far the large voltages (80 V) are problematic for application.

To avoid misunderstandings it should be clearly stated that the observed properties are at 80 K.

As minor comment I would suggest to modify the last sentence of the abstract "These results open up a novel avenue for achieving multifunctional electronic and optoelectronic devices" as it is a very standard sentence written in almost every manuscript and is not bearing particular useful information for the readers.

Figure should also appear in better quality. Some symbols e.g. in Fig 1a, fig. 2b,d

Response to Reviewer #3

We thank you for spending time to review our work and giving your thoughtful comments. Our responses are listed below.

Comment 1: *The manuscript includes detailed experimental characterization and simulation results, which can be useful to better understand these devices and materials but the proposed model for the anomalous transport and its dependence on temperature and illumination is still unclear and confusing. Why would minority carriers (holes) be captured into deep traps at the semiconductor interface under a large positive gate voltage that accumulates the semiconductor interface with electrons? [In fig. 1: "Process I, excess minority carriers (holes) will be captured and stored in Et under non-equilibrium conditions." In fig. 4 Process I, program, ($V_{gs} > 0$).]*

Our reply: The carrier type (electrons or holes) captured by trap states is determined by the position and filling state of the trap states. For bare low-dimensional semiconductors (no hybrid structures), the filling of trap states is involved with the Fermi level, thus n-type channel mainly has hole-trap states while p-type channel mainly has electron-trap states (**Fig. R1a**, *Adv. Sci.* 4, 1700323, 2017). Therefore, although electrons can accumulate in the semiconductor materials when a positive gate voltage is applied, this does not affect the defect levels to capture minority carriers (holes) (**Process I** of **Fig. 4b**). On the one hand, under a large gate voltage, the position of defect states has not changed and is still below the Fermi level. Thus, the trap states have the ability to trap minority carriers (holes). On the other hand, although the device exhibits an undoubted n-type feature, the minority carriers (holes) must exist in the channel material to maintain the charge neutrality.

The **Process I** in **Fig. 1a** (**Fig. R1e**) depicts the schematic band diagrams of capturing minority carriers (holes) in a typical n-type semiconductor with deep traps. Due to the upper Fermi level in n-type semiconductor, the deep traps are filled and can't capture electrons but only capture holes (minority carriers). This fundamental

model (taking into account hole trapping in n-type semiconductors with trap states) has been widely reported and utilized in the previous studies (*Nature Nanotech.* 9, 780, 2014; *Nano Lett.* 14, 6165, 2014; *Adv. Mater.* 30, 1804332, 2018; *Adv. Sci.* 4, 1700323, 2017). The corresponding figures are extracted from the relevant references and displayed in **Fig. R1a-d**. Except for emphasizing the deeper defect levels (causing the robust trap effect), our physical model is completely consistent with them.

Figure R1 | (a-d) Figures extracted from the relevant references (*Adv. Sci.* 4, 1700323, 2017; *Nature Nanotech.* 9, 780, 2014; *Nano Lett.* 14, 6165, 2014; *Adv. Mater.* 30, 1804332, 2018) that describe the band diagram of holes trapping in a n-type

semiconductor with trap states. (e) Our physical model of holes trapping.

Comment 2: *The output characteristics shown in the supplement (S12 and S20) also do not look like those of FETs suggesting a different type of operation involved in these devices, perhaps resulting from specific fabrication conditions. Simpler MOS characterization of this gate stack (Cr/Au//SiO₂//MoSSe) is needed to interpret the FET device I-V characteristics shown here.*

Our reply: The output characteristics (**Fig. S12 and S20**) exhibit the same anomalous temperature-dependent electronic transport and negative photoresponse behaviors as the transfer characteristics of our FETs. As shown in **Fig. S12 (Fig. R2a and R2b)**, the obvious gate tunability is observed at 300 K, while the gate tunability is negligible at 80 K. This phenomenon is completely consistent with the FET's transfer characteristics (**Fig. R2c**).

Figure R2 | (a, b) Output characteristic (I_{DS} - V_{DS}) curves of device #2 under various V_{GS} from -30 V to -70 V. $T = 300$ K and 80 K. (c) Transfer characteristic (I_{DS} - V_{GS}) curves of device #2 at 300 K and 80 K.

Figure S20 (Fig. R3a) shows the I_{DS} - V_{DS} output curves under illumination with 473, 1550 and 1940 nm lasers at 80 K, where negative photoresponse (illumination current is lower than the dark current) is observed at $V_{GS} = -80$ V. And compared with 473 nm laser, the negative photoresponse under infrared lasers (1550 and 1940 nm) is more prominent. These properties are also completely consistent with the FET's transfer characteristics (**Fig. R3b**).

Figure R3 | (a) I_{DS} - V_{DS} output curves and (b) I_{DS} - V_{GS} transfer curves of our device under the dark and illuminated states with 473 nm (248 mW cm^{-2}), 1550 nm ($100.6 \mu\text{W cm}^{-2}$) and 1940 nm ($327.1 \mu\text{W cm}^{-2}$) lasers. $V_{DS} = 1 \text{ V}$, $T = 80 \text{ K}$.

Note that a standard device fabrication process was adopted during the experiment, and no specific fabrication condition was involved. The details about devices fabrication have been provided in the Manuscript (Methods - Device fabrication and characterization). Firstly, few-layer $\text{MoS}_{2x}\text{Se}_{2(1-x)}$ were mechanically peeled and transferred onto 280 nm SiO_2/Si wafers. Electrical contacts were fabricated by standard electron-beam lithography (Nova200 NanoLab) followed by deposition of 10/50-nm-thick Cr/Au electrodes (with a rate of 0.05 \AA s^{-1}). Finally, a standard lift-off process was adopted. These processes have been widely used to fabricate two-dimensional electronic devices (*Nat. Mater.* 12, 815, 2013; *Nano Lett.* 13, 100, 2012; *Nano Lett.* 16, 6383, 2016). To further eliminate the influence of fabrication conditions, MoS_2 device was fabricated with the same process. As shown in **Figure R4**, our MoS_2 device exhibits the general FET properties (the device on/off ratio remains almost constant with the decreasing temperature), suggesting that the anomalous properties of $\text{MoS}_{2x}\text{Se}_{2(1-x)}$ FETs do not result from our fabrication process.

Figure R4 | Control experiments based on few-layer MoS₂ device. (a) AFM images of the MoS₂-based device and the corresponding line profile. (b) I_{DS} - V_{GS} curves under various temperatures. $V_{DS} = 1$ V. The device on/off ratio remains almost constant with the decreasing temperature. (c, d) I_{DS} - V_{DS} curves of the device under various V_{GS} . $T = 300$ K and 80 K.

Response to Reviewer #4

I have carefully read the manuscript by Yin et al. and the latest review round. I find the responses by the authors and the additional experiments sufficiently convincing and can recommend the manuscript for acceptance but need some clarifications

Our reply: We are grateful to the reviewer for the positive evaluation on the manuscript. In the revised manuscript, we have addressed the reviewer's comments and revised our manuscript accordingly. Our responses are listed below.

Comment 1: *In Figure 1d the endurance is provided as 120 cycles. This is quite poor endurance. Could the authors comment on this? In how far the large voltages (80 V) are problematic for application.*

Our reply: For consistency, we consider collecting all the experimental data based on one sample. Therefore, to ensure the safe and normal operation of the device, we only performed 120 program-erase cycles in the first round of testing. Actually, this endurance cycle test is comparable to the reported optoelectronic memories. Examples include MoO_x optoelectronic resistive memory with 12 operation cycles (*Nat. Nanotech.* 2019, <https://doi.org/10.1038/s41565-019-0501-3>), MoTe₂/BN optoelectronic memory with 20 operation cycles (*Adv. Mater.* 30, 1804470, 2018), P3HT/DAE-Me optoelectronic memory with 70 operation cycles (*Nat. Nanotech.* 11, 769, 2016), WSe₂/BN optoelectronic memory with 200 operation cycles (*Nat. Commun.* 9, 2966, 2018), MoS₂/cPVP/gold nanoparticles photoelectronic memory with 200 operation cycles (*Adv. Mater.* 28, 9196, 2016).

As for our devices, it is certain that more endurance cycles can be operated. Based on your comment, as shown in **Fig. R5**, we have performed additional 600 program-erase cycles. The device still maintains its stability and high on/off ratio ($\sim 10^7$), even testing again after a year.

The above discussions about operation endurance and Fig. R5 have been added to page 18 (lines 4-6) in the revised Manuscript and Fig. S26b in the revised Supplementary Information.

Figure R5 | Endurance characteristics of the memory for over 600 program/erase cycles, testing again after a year. Laser wavelength and power density are 1550 nm and $100.6 \mu\text{W cm}^{-2}$, respectively.

As for the operation voltages, it is true that the operation voltages applied in the manuscript are high. This problem is ubiquitous in silicon-based optoelectronic memories. Examples include P3HT/DAE-Me optoelectronic memory operates at $V_g = -60 \text{ V}$ (*Nat. Nanotech.* 11, 769, 2016), WSe₂/BN optoelectronic memory operates at $V_g = 50 \text{ V}$ (*Nat. Commun.* 9, 2966, 2018), MoS₂/SWCNs optoelectronic memory operates at $V_g = -50 \text{ V}$ (*Small* 15, 1804661, 2019). This is mainly because we used 280 nm SiO₂ (its relative dielectric constant is 3.9) as the back-gate dielectric layer. The thick dielectric leads to a poor gate control over the driving current, resulting in high operating voltage and high threshold voltage. Generally, for a capacitance-operated device, the regulating effect of gate voltage on the source-drain current is closely related to gate capacitance, which is most simply expressed as: $C = \epsilon_0 K A / t$, where ϵ_0 is the permittivity of free space, K is the relative dielectric constant, A is the area and t is the dielectric thickness (*Materials Science and Engineering: R: Reports* 2015, 88, 1). Thus, (i) thinner dielectric layer can provide greater gate control over the field-effect device; (ii) having a higher dielectric constant means the gate insulator can provide increased capacitance between two conducting plates—storing more charge—for the same thickness of insulator (*Adv. Mater.* 26, 6255, 2014; *IEEE*

spectrum 44, 29, 2007).

We have demonstrated that the robust trap effect is independent of the substrate (**Fig. S16**), so the device operation voltage can be effectively reduced by introducing high-*k* dielectric layer. Based on your comments, the above discussion has been added to page 20 (lines 3-5) of the revised Manuscript.

Comment 2: *To avoid misunderstandings it should be clearly stated that the observed properties are at 80 K.*

Our reply: We have clearly stated the test temperature. Please see the marked text in the Introduction section (page 3, line 15 in our revised Manuscript).

Comment 3: *As minor comment I would suggest to modify the last sentence of the abstract “These results open up a novel avenue for achieving multifunctional electronic and optoelectronic devices” as it is a very standard sentence written in almost every manuscript and is not bearing particular useful information for the readers.*

Our reply: We have modified the sentence as “This demonstration of defect engineering in two-dimensional semiconductors opens up a novel avenue for achieving high-performance infrared detector and nonvolatile optoelectronic memory.” Please see the marked text on page 2, lines 4-7 in our revised Manuscript.

Comment 4: *Figure should also appear in better quality. Some symbols e.g. in Fig 1a, fig. 2b,d*

Our reply: Many thanks for this kindly reminder. We have re-plotted the figures to make it easy to read. Please see the new figures (**Fig. R6-8** below) in the revised Manuscript.

Figure R6 | The revised Fig. 1.

Figure R7 | The revised Fig. 2.

Figure R8 | The revised Fig. 4.

REVIEWERS' COMMENTS:

Reviewer #4 (Remarks to the Author):

The authors have addressed the comments and amended the manuscript. In my opinion it can be accepted for publication.

Response to Reviewer #4

The authors have addressed the comments and amended the manuscript. In my opinion it can be accepted for publication.

Our reply: We would like to thank you for reviewing our paper, we appreciate your insightful comments on our research.